# Allosteric inhibition of the T cell receptor by a designed membrane ligand

Yujie Ye[1], Shumpei Morita[2], Justin J Chang[3], Patrick M Buckley[4], Kiera B Wilhelm[2], Daniel DiMaio[3], Jay T Groves[2,5], Francisco N Barrera[1]*

[1]Department of Biochemistry & Cellular and Molecular Biology, University of Tennessee at Knoxville, Knoxville, United States; [2]Department of Chemistry, University of California, Berkeley, Berkeley, United States; [3]Department of Genetics, Yale University, New Haven, United States; [4]Department of Microbial Pathogenesis, Yale University, New Haven, United States; [5]Institute for Digital Molecular Analytics and Science, Nanyang Technological University, Singapore, Singapore

*For correspondence:
fbarrera@utk.edu

Competing interest: The authors declare that no competing interests exist.

**Abstract** The T cell receptor (TCR) is a complex molecular machine that directs the activation of T cells, allowing the immune system to fight pathogens and cancer cells. Despite decades of investigation, the molecular mechanism of TCR activation is still controversial. One of the leading activation hypotheses is the allosteric model. This model posits that binding of pMHC at the extracellular domain triggers a dynamic change in the transmembrane (TM) domain of the TCR subunits, which leads to signaling at the cytoplasmic side. We sought to test this hypothesis by creating a TM ligand for TCR. Previously we described a method to create a soluble peptide capable of inserting into membranes and binding to the TM domain of the receptor tyrosine kinase EphA2 (Alves et al., eLife, 2018). Here, we show that the approach is generalizable to complex membrane receptors, by designing a TM ligand for TCR. We observed that the designed peptide caused a reduction of Lck phosphorylation of TCR at the CD3 ζ subunit in T cells. As a result, in the presence of this peptide inhibitor of TCR (PITCR), the proximal signaling cascade downstream of TCR activation was significantly dampened. Co-localization and co-immunoprecipitation in diisobutylene maleic acid (DIBMA) native nanodiscs confirmed that PITCR was able to bind to the TCR. AlphaFold-Multimer predicted that PITCR binds to the TM region of TCR, where it interacts with the two CD3 ζ subunits. Our results additionally indicate that PITCR disrupts the allosteric changes in the compactness of the TM bundle that occur upon TCR activation, lending support to the allosteric TCR activation model. The TCR inhibition achieved by PITCR might be useful to treat inflammatory and autoimmune diseases and to prevent organ transplant rejection, as in these conditions aberrant activation of TCR contributes to disease.

## Editor's evaluation

The authors combine AlphaFold-Multimer and a previously described technology of designing soluble transmembrane-targeting peptides that interfere with the function of the T cell receptor (TCR). This study provides important insights into the molecular mechanism of T cell activation. The approach is convincing since the designed PITCR peptide has functional effects, in contrast with the predicted negative controls PITCRG41P and pHLIP. The results and the methods from this study will be of interest to those studying the TCR, as well as those seeking to use the TCR or its derivatives in synthetic biology studies and immunotherapy.

## Introduction

T cells are central players in the adaptive immune response. Different types of T cells recognize the presence of pathogenic organisms and cancer cells and orchestrate diverse immune activities intended to kill the damaging cells (*Courtney et al., 2017*; *Ganti et al., 2020*). The T cell receptor (TCR) is a protein complex present at the membrane of T cells that allows detection of foreign molecules. The TCR engages with antigen-presenting cells (APCs), where peptide fragments are displayed at the major histocompatibility complex (pMHC) (*Chakraborty and Weiss, 2014*). Recognition of pMHC by the TCR triggers an intricate signaling cascade that activates the T cell response (*Courtney et al., 2018*; *Kuhns and Davis, 2008*). In αβ T cells, pMHC binding occurs at the TCRαβ subunits. The TCR complex additionally contains four types of CD3 subunits: ε, γ, δ, and ζ, forming the TCR-CD3 complex (referred herein as TCR). The dominant stoichiometry of TCR is composed of TCRαβ-CD3εγ-CD3εδ-CD3ζζ (*Call et al., 2002*; *Dong et al., 2019*; *Mariuzza et al., 2020*). The CD3 subunits relay the information of the pMHC binding event across the membrane, initiating TCR proximal signaling. The TCR downstream signaling cascade starts by phosphorylation of ITAMs (immunoreceptor tyrosine-based activation motifs) present in all CD3 subunits, particularly at ζ, which leads to a transient increase in calcium levels in the cytoplasm (*Au-Yeung et al., 2018*; *Lo et al., 2018*).

Despite decades of effort, there is not yet a clear understanding of how TCR is activated after recognition of pMHC (*Mariuzza et al., 2020*). A number of different activation modes have been proposed, namely the aggregation/clustering model, the segregation model, the mechanosensing model, and the allosteric model. In this latter functional hypothesis, the signal resulting from binding of pMHC to the extracellular region of the TCRαβ is dynamically transmitted across the transmembrane (TM) region into the ITAMs. While there is growing evidence supporting the allosteric model (*Chen et al., 2022*; *Schamel et al., 2019*), the molecular mechanism of the allosteric triggering of TCR-CD3 is not clear. Recent reports provide a plausible mechanistic framework on how allosteric changes affect the TM helical bundle: TCR activation involves a quaternary relaxation of the TM helical bundle, whereby in the active state there is a loosening in the interaction between the TM helices of the TCRβ and CDζ (*Lanz et al., 2021*; *Prakaash et al., 2021*).

Here, we have used a rational design approach to develop a peptide (PITCR) to target the TM region of TCR. PITCR comprises the TM domain of the ζ subunit (*Call et al., 2006*) modified by the addition of acidic residues to convert it to a conditional TM sequence. The PITCR peptide is used to test the allosteric relaxation model, as its binding to the TM region of TCR can be reasonably expected to alter the conformation and/or dynamics/packing of the helical bundle. We observed that PITCR robustly inhibited the activation of the TCR. The results obtained in this work support the allosteric relaxation activation model and provide new mechanistic insights into TCR activation.

## Results

### PITCR decreases phosphorylation of the ζ chain upon TCR activation

We recently reported an approach to transform the isolated TM domains of human receptors into peptides that function as conditional TM sequences: they are highly soluble in water, while they have the ability to insert into the membrane in the TM orientation that allows the peptide to interact laterally with their natural binding partners (*Alves et al., 2018*). We applied this approach to the CD3ζ TM, to generate the PITCR peptide. To this end we introduced glutamic acid residues in positions which are not expected to hinder interactions with other TCR subunits, according to cryoEM structures of the TCR.

Biophysical experiments in synthetic lipid vesicles showed that the design for PITCR was successful, as it was soluble in aqueous solution and able to insert into membranes (*Figure 1—figure supplement 1*).

The TCR at the surface of human Jurkat T cells is activated upon binding of the monoclonal antibody (mAb) OKT3, which has been widely applied to study T cell signaling (*Lo et al., 2018*; *Lo et al., 2019*). TCR activation is initiated by phosphorylation of tyrosine residues at the ITAMs of the ζ chain by Lck (lymphocyte-specific protein tyrosine kinase) (*Figure 1A*; *Courtney et al., 2017*). To investigate whether PITCR affected TCR activation, we treated Jurkat cells with PITCR before stimulation with OKT3. The immunoblot results revealed that PITCR reduced phosphorylation of the ζ chain at

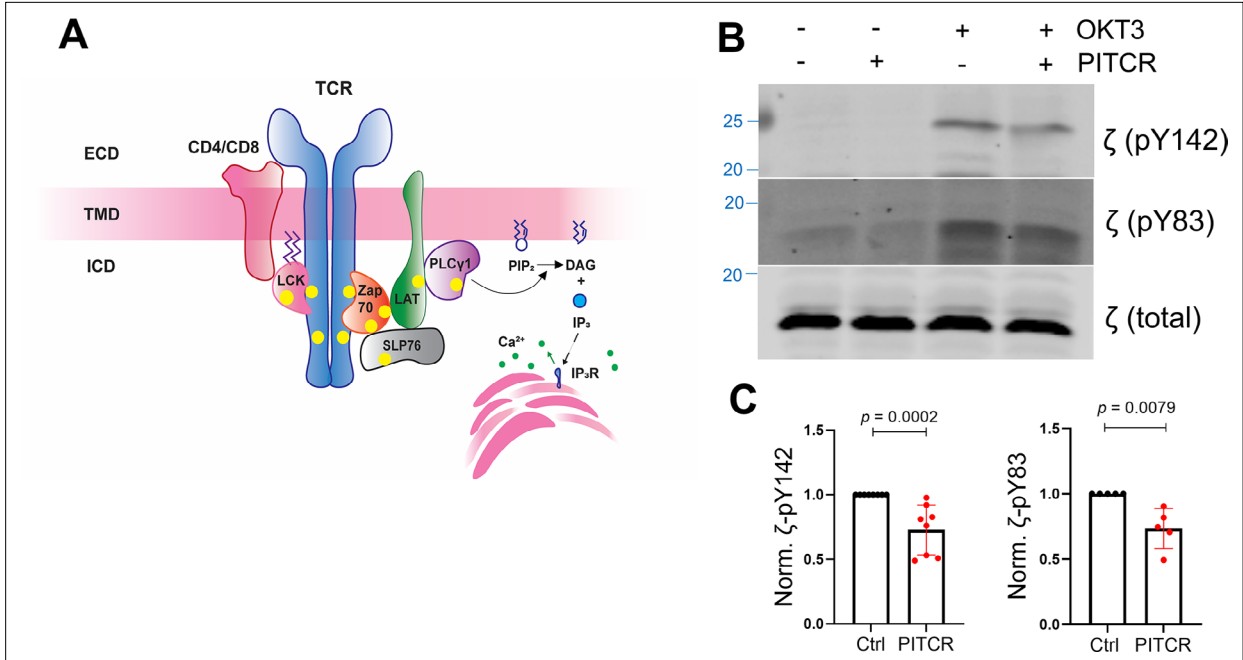

**Figure 1.** Peptide inhibitor of T cell receptor (PITCR) reduces phosphorylation of the ζ chain in response to OKT3. (**A**) Cartoon that illustrates TCR proximal downstream signaling. The plasma membrane is shown as a horizontal bar, and phosphorylation sites are shown as yellow dots. ECD: extracellular domain; TMD: transmembrane domain; ICD: intracellular domain of TCR. (**B**) Jurkat cells were treated with PITCR, followed by stimulation with OKT3. Lysates were analyzed by immunoblot to detect TCR phosphorylation of ζ (pY142 and pY83). Total ζ levels were assessed and no change was observed. Data are representative of at least five independent experiments. (**C**) Quantification of phosphorylation at both tyrosine residues in the presence of OKT3, normalized to data in the absence of PITCR (based on data from *Figure 1—figure supplement 2*). Error bars are the SD. p-Values were calculated using a two-tailed Mann-Whitney test.

The online version of this article includes the following source data and figure supplement(s) for figure 1:

**Source data 1.** Original western blots.

**Figure supplement 1.** The secondary structures of PITCR and PITCRG41P at a physiological pH and an acidic pH.

**Figure supplement 2.** Quantification of phosphorylation of ζ (pY142) (**A**) and ζ (pY83) (**B**) after OKT3 stimulation.

residues Y142 and Y83 after TCR activation (*Figure 1B–C* and *Figure 1—figure supplement 2*). These results suggest that PITCR reduces activation of the TCR.

## Phosphorylation of TCR proximal signaling molecules is downregulated by PITCR

Since PITCR inhibited TCR phosphorylation after activation, we sought to explore the effect of PITCR on TCR downstream signaling. TCR activation induces the recruitment of Zap70 (ζ chain-associated protein kinase 70) to the phosphorylated TCR, where it is itself phosphorylated by Lck (*Figure 1A*). The activated Zap70 then phosphorylates LAT (linker for activation of T cells) and SLP76 (SH2 domain containing leukocyte protein of 76 kDa), and as a result PLCγ1 (phospholipase C-γ1) is recruited and phosphorylated (*Courtney et al., 2018*; *Lo et al., 2018*; *Lo and Weiss, 2021*). In agreement with the hypothesis that PITCR inhibits TCR activation, in the presence of peptide we observed a statistically significant decrease in the phosphorylation of Zap70, LAT, SLP76, and PLCγ1 (*Figure 2* and *Figure 2—figure supplement 1*). However, PITCR did not affect phosphorylation of Lck (*Figure 2—figure supplement 2*). Since basal phosphorylation of Zap70, LAT, SLP76, and PLCγ1 was observed in the absence of OKT3 (*Figure 2A*), the immunoblot results indicate that the effect of PITCR is specific to stimulation of the TCR. Our data therefore indicate that PITCR causes a robust inhibition of the proximal signaling cascade triggered by TCR activation.

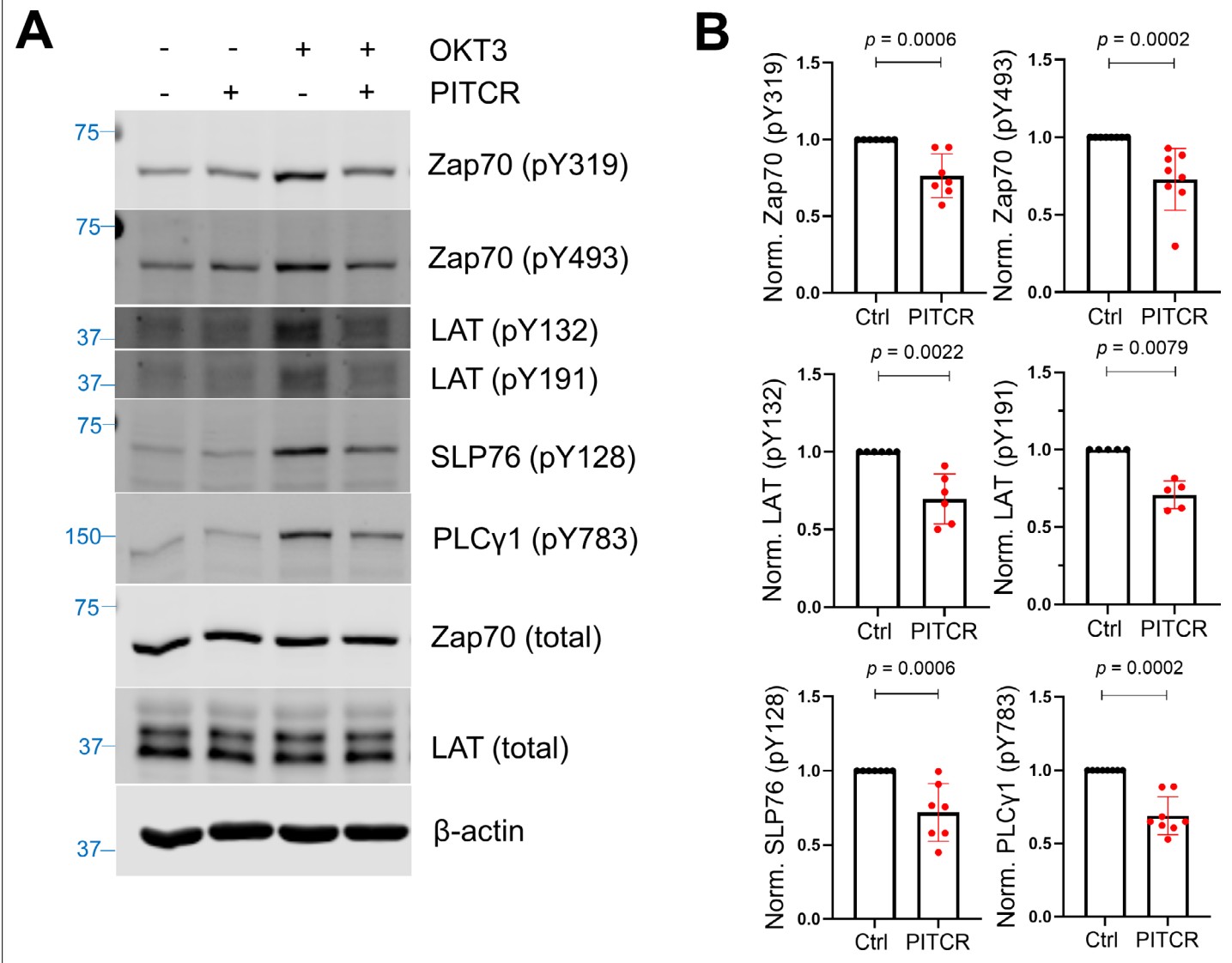

**Figure 2.** Peptide inhibitor of T cell receptor (PITCR) reduces phosphorylation of TCR proximal signaling proteins after activation. (**A**) Immunoblot analysis of lysates to detect phosphorylation of Zap70 (pY319 and pY493), LAT (pY132 and pY191), SLP76 (pY128), and PLCγ1 (pY783). Total protein levels of Zap70, LAT, and the housekeeping protein β-actin were assessed, revealing no changes. Data are representative of at least five independent experiments. (**B**) Quantification of phosphorylation in the presence of OKT3, normalized to data in the absence of PITCR (based on data from *Figure 2—figure supplement 1*). Error bars are the SD. p-Values were calculated using a two-tailed Mann-Whitney test.

The online version of this article includes the following source data and figure supplement(s) for figure 2:

**Source data 1.** Quantification results of multiple TCR proximal phophorylated proteins.

**Figure supplement 1.** Quantification of Zap70 (pY319), Zap70 (pY493), LAT (pY132), LAT (pY191), SLP76 (pY128), and PLCγ1 (pY783) in response to OKT3 stimulation.

**Figure supplement 2.** Peptide inhibitor of T cell receptor (PITCR) does not reduce phosphorylation of Lck after OKT3 activation.

**Figure supplement 2—source data 1.** Quantification results of phosphorylated Lck in presence of PITCR.

## PITCR reduces the intracellular calcium response

After TCR activation, the active PLCγ1 hydrolyzes phosphatidyl inositol 4,5-bisphosphate to generate inositol trisphosphate ($IP_3$) and diacylglycerol. Free $IP_3$ diffuses across the cytoplasm and binds to the $IP_3$ receptor at the endoplasmic reticulum (ER), causing the release of calcium from ER storage (*Figure 1A*; *Courtney et al., 2018*; *Lewis, 2001*; *Trebak and Kinet, 2019*). Based on our previous results, we reasoned that PITCR should inhibit the cytoplasmic calcium influx that follows TCR

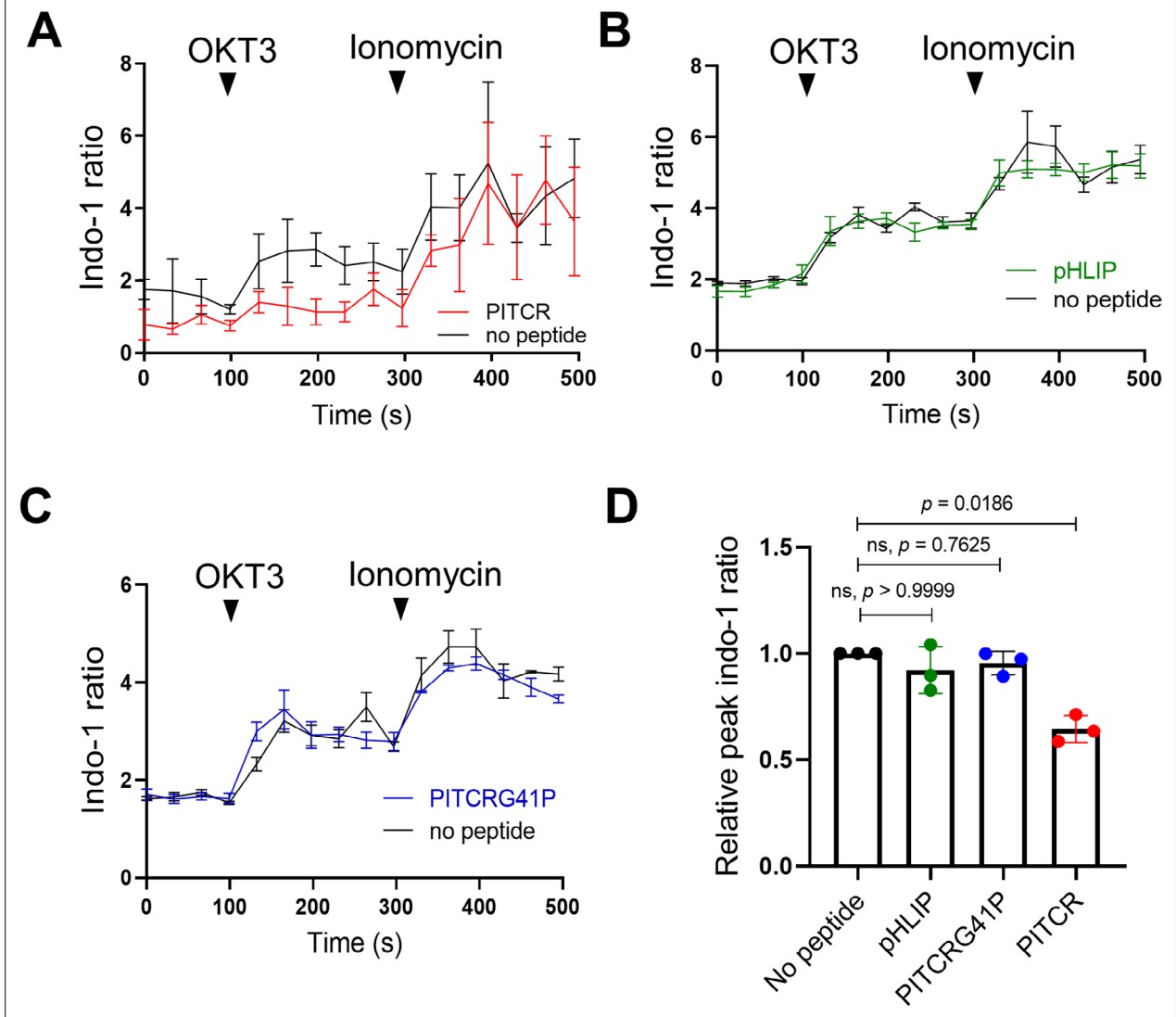

**Figure 3.** Peptide inhibitor of T cell receptor (PITCR) reduces the TCR intracellular calcium response. Jurkat cells were stained with the fluorescent dye Indo-1 AM, followed by treatment with PITCR (**A**), pHLIP as a negative control (**B**), or the variant PITCRG41P (**C**) and stimulated with OKT3. Ionomycin was applied as a positive calcium control. The Indo-1 ratio was calculated from fluorescence at 405 nm (calcium-bound) divided by 475 nm (calcium-free). Data are representative of three independent experiments. Each independent experiment includes at least four technical replicates. Error bars are the SEM. (**D**) Quantification of the maximum Indo-1 increase after OKT3 activation, normalized to no peptide treatment. Error bars are the SD. p-Values were calculated from a Kruskal-Wallis test with Dunn's multiple comparisons test.

The online version of this article includes the following figure supplement(s) for figure 3:

**Figure supplement 1.** Peptide inhibitor of T cell receptor (PITCR) reduces intracellular calcium responses for PITCR (**A**), but not for pHLIP (**B**) or PITCRG41P (**C**).

activation. We tested this idea with a kinetic analysis of intracellular calcium release using the calcium indicator Indo-1 (*Lo et al., 2018*).

Consistent with our expectation, we observed that the calcium response after OKT3 stimulation was attenuated in the PITCR treatment group compared with control conditions (*Figure 3A and D*, *Figure 3—figure supplement 1*). We used as a negative control pHLIP (*Scott et al., 2019*; *Scott et al., 2017*), a different conditional TM peptide that is not expected to interact with membrane proteins (*Alves et al., 2018*). The dynamic calcium curve of the pHLIP-treated group was within the error of the control curve (*Figure 3B and D*). To further test the specificity of PITCR, we performed a mutation (replacing Gly41 for a Pro) in PITCR that is expected to form a helical kink and disrupt the TM helix. Control biophysical experiments indicated that the G41P mutation did not prevent the

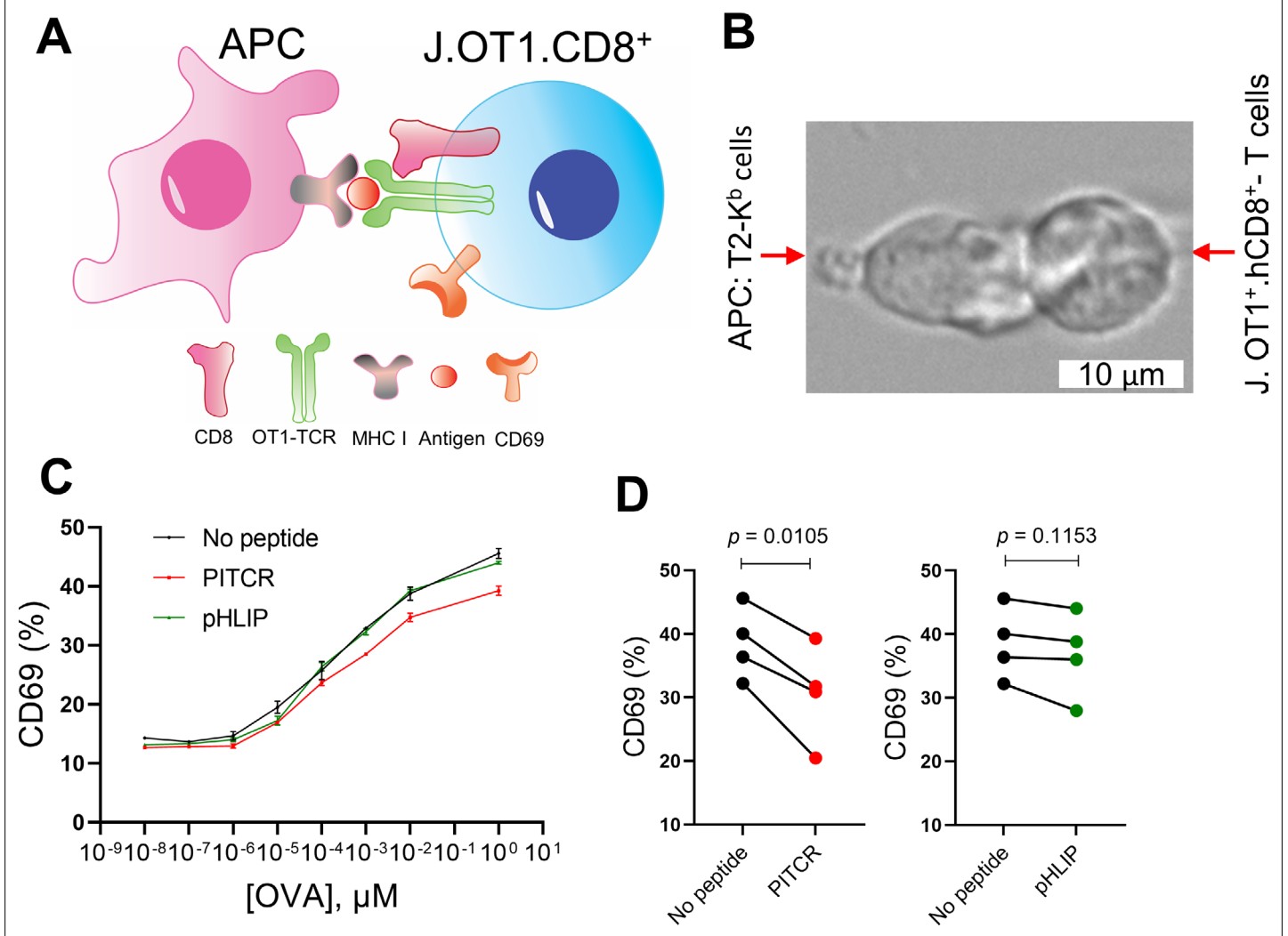

**Figure 4.** Peptide inhibitor of T cell receptor (PITCR) reduces CD69 expression after T cell activation by antigen-presenting cell (APC). (**A**) Cartoon showing T cell interaction with APC. (**B**) A live cell microscopy image that depicts engineered Jurkat-OT1[+] TCR-CD8[+] T cells interacting with T2Kb APC pre-incubated with the peptide antigen ovalbumin (OVA). (**C**) Jurkat-OT1[+] TCR-CD8[+] T cells were treated with PITCR or pHLIP (as a negative control), and then incubated with T2Kb cells at different concentrations of OVA, followed by CD69 flow cytometry analysis. The upregulation of CD69 is representative of four independent experiments. Each independent experiment includes two technical replicates. Error bars are the SD. (**D**) Quantification of CD69-positive cells at [OVA]=1 µM for PITCR (red) and negative control pHLIP (green). Each dot pair represents one independent experiment. p-Values were calculated from two-tailed paired t-test.

peptide to act as a conditional TM (*Figure 1—figure supplement 1*). We observed that PITCRG41P was unable to inhibit the calcium influx in response to OKT3 treatment (*Figure 3C and D*). These data indicate that PITCR specifically impaired the calcium response that occurs in Jurkat cells after TCR activation, in agreement with the observed decrease in phosphorylation of PLCγ1 (*Figure 2A*).

## Inhibition of TCR activation by APCs

While the OKT3 mAb efficiently stimulates TCR signaling, we sought to test the effect of TCR in a more physiologically relevant TCR system, consisting of T cell stimulation by binding to pMHC in APCs (*Lo et al., 2018*). Similar to human cytotoxic CD8[+] T cells, OT1[+]-TCR CD8[+] Jurkat T cells (J.OT1.CD8) can recognize the ovalbumin (OVA) peptide presented by H-2K[b]-MHC I on T2 APCs (T2K[b]) (*Figure 4A and B*). TCR engagement results in increased levels of CD69, a T cell activation marker (*Lo et al., 2018*; *Lo et al., 2019*; *Wolpert et al., 1997*). We treated J.OT1.CD8 cells with PITCR followed by incubation with T2K[b] cells pre-treated with a range of OVA concentrations and measured the CD69 expression using flow cytometry (*Figure 4C and D*). As expected, in the presence of high OVA concentrations we

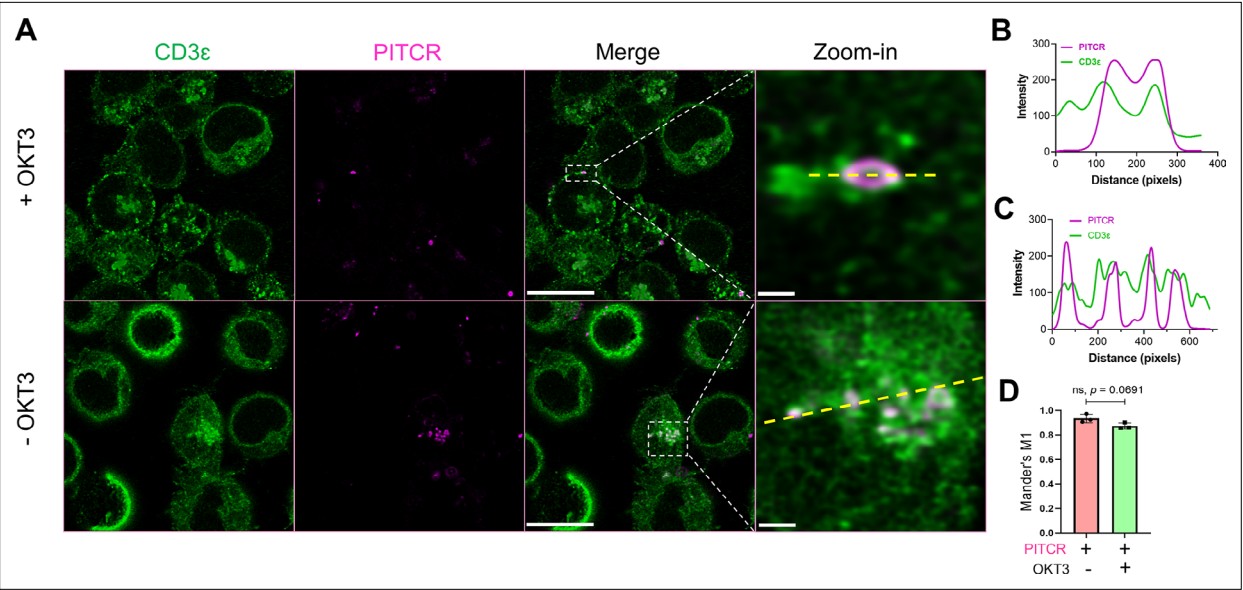

**Figure 5.** Peptide inhibitor of T cell receptor (PITCR) co-localizes with TCR. (**A**) PITCR680 and CD3ε co-localization was studied in the presence and absence of OKT3. Scale bars = 10 µm. Representative areas with co-localization at the plasma membrane (*top*) and cytoplasm (*bottom*) were zoomed-in, where scale bars are 0.5 µm and 1 µm, respectively. Confocal images are representative of three independent experiments. (**B**) and (**C**) show graphic profile curves (dotted yellow lines) plotted across the zoom-in regions of interest (ROI) in +OKT3 and –OKT3, respectively. Magenta lines denote PITCR, and green lines denote CD3ε. Overlap indicates co-localization. (**D**) Quantification of co-localization by Mander's coefficient (**M1**), corresponding to the fraction of PITCR that overlaps with CD3ε. Error bars indicate SD. p-Value was calculated from two-tailed unpaired t-test.

The online version of this article includes the following figure supplement(s) for figure 5:

**Figure supplement 1.** Peptide inhibitor of T cell receptor (PITCR) co-localizes with TCR.

**Figure supplement 2.** Matrix-assisted laser desorption ionization-time-of-flight (MALDI-TOF) spectra of NEC-peptide inhibitor of T cell receptor (PITCR).

observed CD69 upregulation (*Figure 4C*). Our data showed that PITCR caused a significant reduction of CD69 levels in response to 1 µM OVA stimulation (*Figure 4C and D*). To further examine the specificity of PITCR, we again used pHLIP as a negative control, and we observed that pHLIP did not elicit significant changes. Our data therefore indicate that PITCR specifically impaired CD69 upregulation in J.OT1.CD8 cells in response to OVA stimulation. These results show that PITCR also achieves robust inhibition when TCR is activated by binding to pMHC presented by APC.

## PITCR co-localizes with the TCR in Jurkat T cells

Our results in Jurkat cells clearly show that PITCR reduces TCR activation. We sought next to determine if this was a specific effect that resulted from a direct interaction between the peptide and TCR. First, we performed co-localization experiments in Jurkat cells. To this end, we fluorescently labeled PITCR with dylight680 (PITCR680). We employed super-resolution confocal microscopy with lightning deconvolution to investigate PITCR680 co-localization with TCR, as detected with an anti-CD3 ε antibody. We observed that PITCR680 localized to some areas of the Jurkat cells, corresponding to the plasma membrane and intracellular organelles, probably endosomes (*Figure 5A*). In these two regions we observed co-localization between PITCR680 (magenta) and TCR (CD3ε, green) (*Figure 5A*). To better assess co-localization, we plotted graphic profile curves on regions of interest, which revealed clear overlap in some areas. To quantify the degree of co-localization, we calculated the Mander's M1 coefficient, which showed a value of ~0.8 (*Figure 5D*). This result reveals that around 80% of PITCR680 signal overlaps with the TCR. We also observed robust co-localization using the Pearson's correlation coefficient ($r$ = ~0.4) (*Figure 5—figure supplement 2*; *Costes et al., 2004*). To further explore whether TCR activation influences co-localization, we stimulated PITCR680-treated Jurkat cells with OKT3. While co-localization results are not proof of interaction, they suggest that PITCR is able to bind to TCR before it is activated by OKT3.

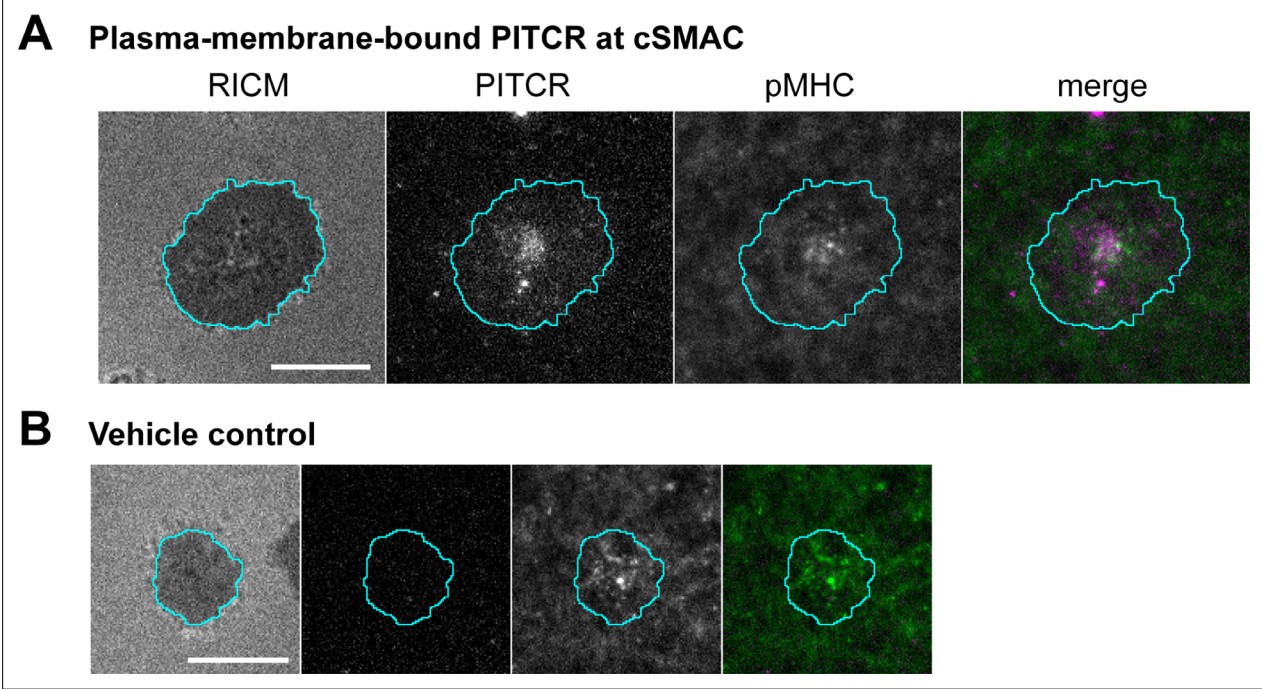

**Figure 6.** Co-localization of peptide inhibitor of T cell receptor (PITCR) and the TCR-pMHC complex in primary murine CD4+ T cells. (**A**) Images of plasma-membrane-localized PITCR555 and TCR-pMHC complex in a representative T cell adhering to supported lipid bilayer functionalized with pMHC (19–23 molecules/μm², 9% labeled with Atto-647N) and ICAM-1 (~20 molecules/μm²). TCR-pMHC complex was selectively visualized with a long exposure time (500 ms). PITCR exhibited localization at central supramolecular activation cluster (c-SMAC) together with the TCR-pMHC complex. (**B**) Vehicle control showed no signal at the PITCR channel. Panel labels correspond to those in A.

The online version of this article includes the following figure supplement(s) for figure 6:

**Figure supplement 1.** The NFAT dose-response curve of primary murine T cells is unaffected by peptide inhibitor of T cell receptor (PITCR).

### Co-localization of PITCR with ligand-bound TCR in primary murine cells

Primary murine CD4+ T cells provided an orthogonal method to assess co-localization of the peptide with TCR. Splenocytes from the TCR(AND) mice, hemizygous for H2k, were pulsed with 1 μM moth cytochrome *c* (MCC) peptide and cultured with the T cells for 2 days. The T cell blasts were treated with IL-2 from the day after harvest to the fifth day after harvest, at which point the cells were used in experiments. T cells in this state respond to antigen with single-molecule sensitivity. T cells treated with either PITCR or a vehicle control were stimulated by contact with supported bilayers functionalized with agonist pMHC (MCC peptide labeled with Atto647N) and the adhesion molecule ICAM-1 (*Lin et al., 2019*; *McAffee et al., 2021*). The primary mouse T cells activated normally upon treatment with PITCR as measured by NFAT translocation (*Figure 6—figure supplement 1*; *Lin et al., 2019*). We performed surface-selective imaging by total internal reflection fluorescence microscopy and immune synapse formation was imaged (*Biswas and Groves, 2019*; *Grakoui et al., 1999*; *Mossman et al., 2005*; *Yu et al., 2012*). TCR-pMHC complexes were selectively distinguished from free pMHC using an elongated image exposure time strategy (*Lin et al., 2019*; *O'Donoghue et al., 2013*; *Pielak et al., 2017*). We observed the c-SMAC (central supramolecular activation cluster) structure (*Figure 6*), as previously reported for activated T cells (*Bromley et al., 2001*; *Grakoui et al., 1999*). PITCR conjugated to AZDye 555 (PITCR555) could be detected in

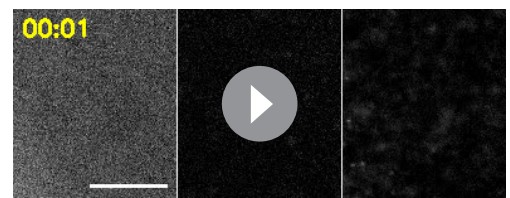

**Video 1.** Real-time imaging of peptide inhibitor of T cell receptor (PITCR) and pMHC in a T cell adhering to supported bilayer. Left: RICM, center: PITCR555, right: pMHC. Cell footprint is shown as cyan line. Scale bar: 10 μm.

https://elifesciences.org/articles/82861/figures#video1

intracellular structures (*Video 1*), consistent with the confocal microscopy results. Additionally, a distinct population of plasma-membrane-bound peptide could be also observed in some cells (*Figure 6*). In these cases, PITCR exhibited c-SMAC localization together with TCR-pMHC complexes. Although the image resolution was insufficient to definitively confirm molecular binding between PITCR and the TCR-pMHC complex, their co-localization is suggestive of interaction. We note that PITCR is a partial TCR inhibitor, and thus is not expected to significantly block c-SMAC formation or activation in these primed mouse T cells due to their extreme sensitivity to antigen. Additionally, TM-TM interactions are often highly sensitive to mutations (*He et al., 2017*; *Westerfield et al., 2021*). Several TM residues in the CD3 subunits that according to our model interact with TCR are different in the murine and the human amino acid sequences. Therefore, we expect PITCR to be less efficient targeting the mouse TCR. In aggregate, the co-localization experiments support binding of PITCR to TCR in human Jurkat T cells and mouse primary T cells.

## PITCR interacts with the TCR in Jurkat T cells

We sought to determine if the observed co-localization indeed resulted from binding between PITCR and TCR. We developed a new assay to maintain the TCR complex of Jurkat cells in a native lipid

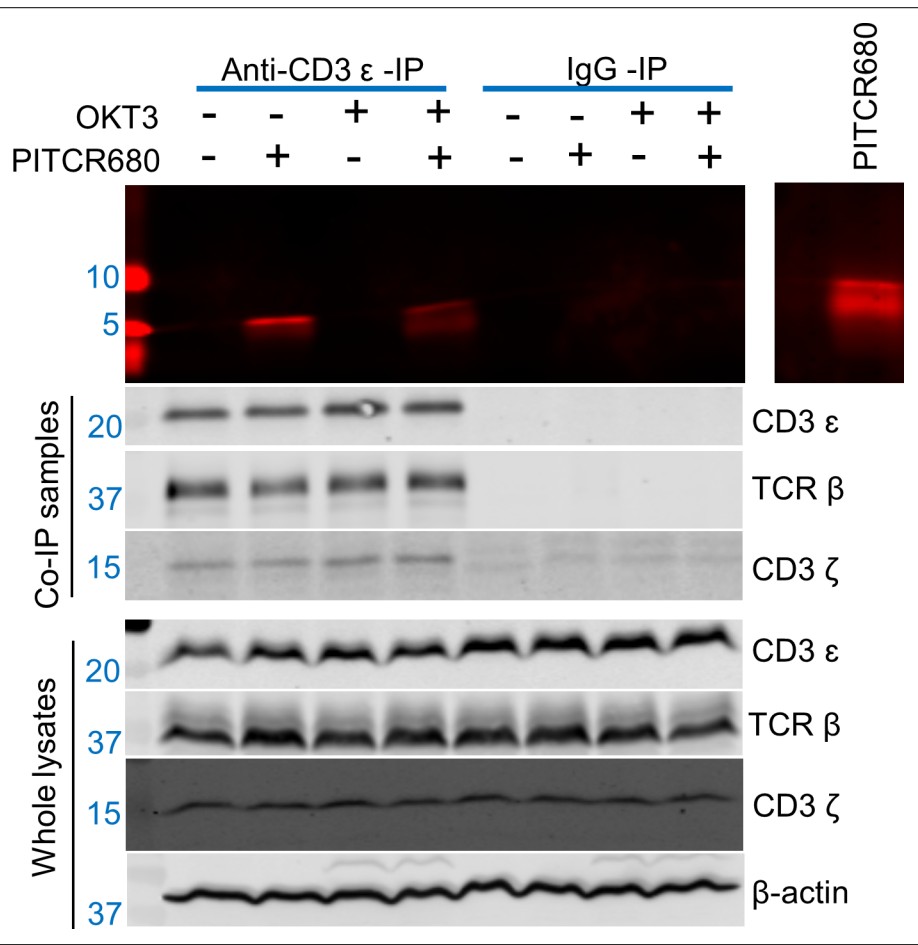

**Figure 7.** Peptide inhibitor of T cell receptor (PITCR) interacts with TCR. Jurkat cells were treated with PITCR-680. TCR nanodiscs were immunoprecipitated with the monoclonal antibody (mAb) anti-CD3 (UCHT1 clone), or with IgG as a negative control. Fluorescent detection of PITCR-680 after co-immunoprecipitation (Co-IP) or run in the gel directly as a positive control (*right side panel*). Below are shown immunoblot analysis of Co-IP samples and whole lysates to probe CD3ε, TCRβ, and CD3 ζ . β-Actin was a loading control. Data are representative of at least three independent experiments.

The online version of this article includes the following source data for figure 7:

**Source data 1.** This data contains the PITCR immunoprecipitation results.

environment, consisting of using the polymer diisobutylene maleic acid (DIBMA) to form native nanodiscs. On these samples we performed a co-immunoprecipitation (Co-IP) experiment using an anti-CD3ε antibody (UCHT1). We observed bands of TCRβ, CD3ε, and CD3ζ in the anti-CD3ε Co-IP lysates (*Figure 7*), indicating that TCR had been successfully immunoprecipitated after solubilization with DIBMA. When cells were incubated with PITCR680, we observed in the Co-IP samples a fluorescent band of molecular weight (~5 kDa) similar to that of PITCR680 (4.8 kDa) (*Figure 7*). To further explore whether TCR activation would affect binding of PITCR680 to the complex, we applied OKT3 as previously described. We observed the ~5 kDa fluorescent band as well. These results indicate that PITCR680 interacts with the TCR irrespective of activation by OKT3, in agreement with the co-localization results in Jurkat cells (*Figure 5*).

## PITCR weakens the interaction of the ζ subunit with the rest of the complex after TCR activation

We next sought to understand the mechanism by which PITCR partially inhibits TCR activation. It has been recently proposed that TCR activation involves a change in robustness of the TM helical bundle. This allosteric change can be detected by immunoprecipitation after solubilization in DDM, as a loose complex is less resistant to this detergent (*Lanz et al., 2021*; *Prakaash et al., 2021*). We optimized this assay for our experimental conditions and investigated the interaction between the ζ chain and rest of the TCR complex. We first examined the immunoprecipitated levels of CD3ε, CD3ζ, and TCRβ

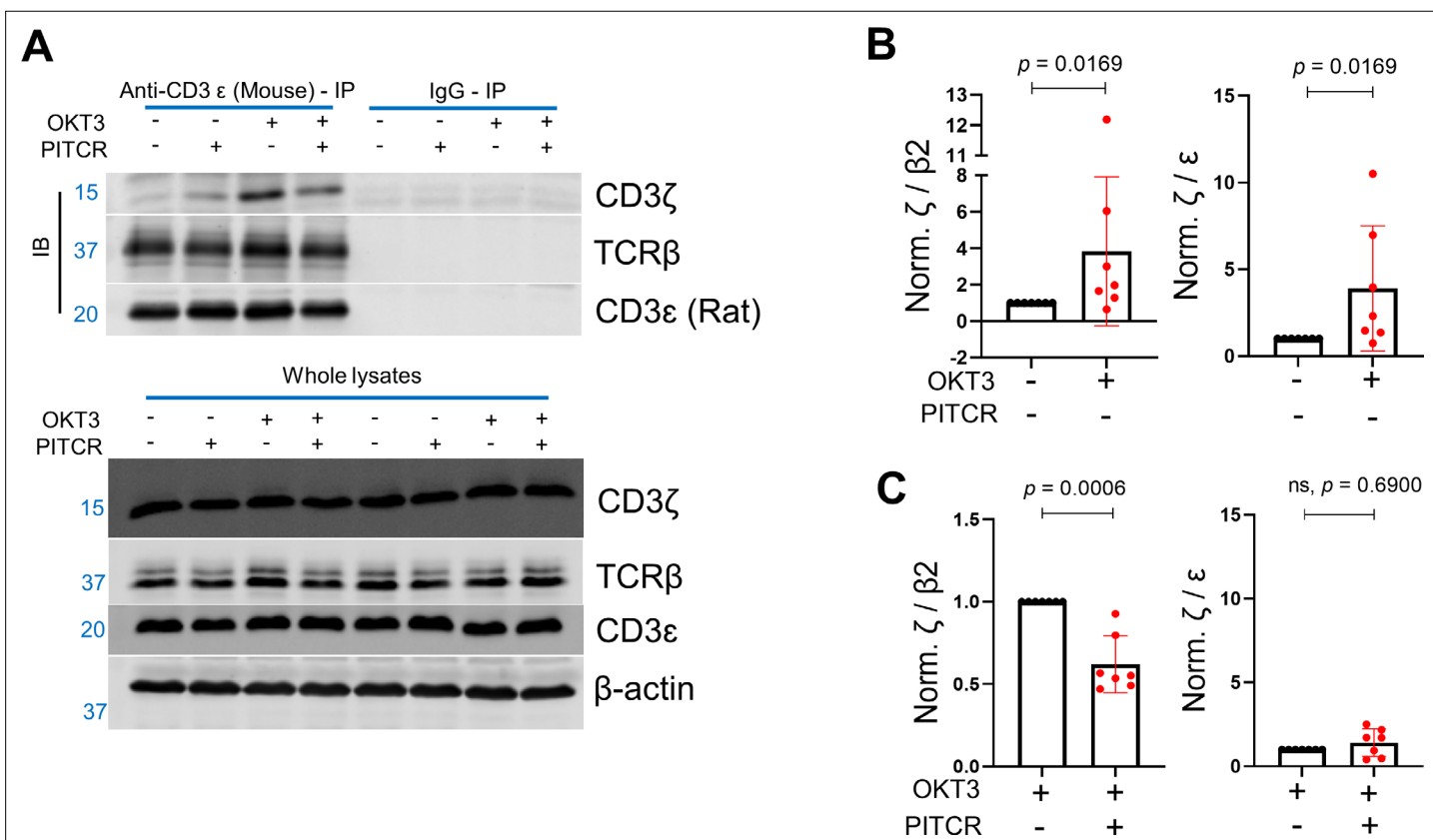

**Figure 8.** Peptide inhibitor of T cell receptor (PITCR) weakens the interaction of the ζ chain with the rest of the complex after TCR activation. (**A**) Immunoblot analysis of immunoprecipitated samples and whole lysate samples solubilized with DDM. Data are representative of at least three independent experiments. (**B**) Quantification of ζ/β2 and ζ/ε after OKT3 stimulation. (**C**) ζ/β2 and ζ/ε in PITCR-treated OKT3 samples, normalized to OKT3 stimulation. The β2 subunit was studied since it is the most abundant β subunit at the plasma membrane. Error bars are SD. p-Values were calculated from two-tailed Mann-Whitney test.

The online version of this article includes the following source data and figure supplement(s) for figure 8:

**Source data 1.** Quantification of immunoprecipitation results in presence of PITCR with/without OKT3 stimulations.

**Figure supplement 1.** Quantification of DDM immunoprecipitation results for the following conditions.

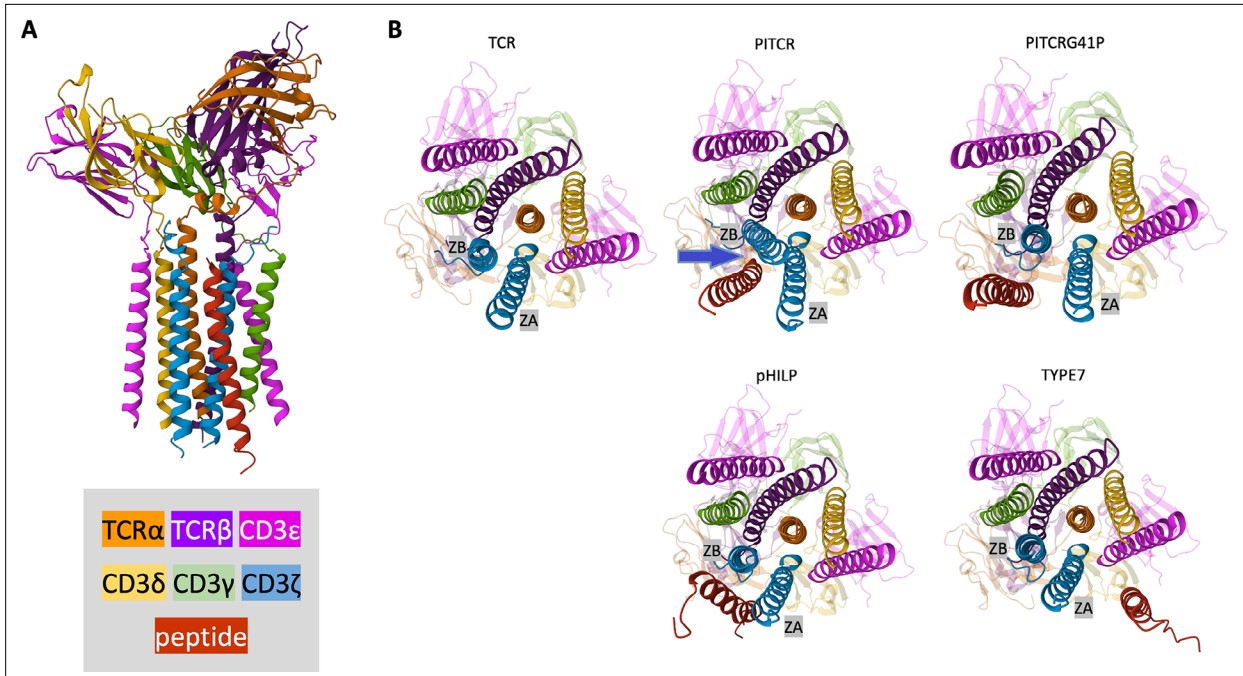

**Figure 9.** AlphaFold-Multimer (AlFoM) model for peptide binding to T cell receptor (TCR). (**A**) Side view of the model that shows the TCR bound to peptide inhibitor of TCR (PITCR) (red). The N-terminus of PITCR is at the top. TCR subunits are colored as shown in the legend. (**B**) Bottom (cytoplasmic) view of the isolated TCR is shown as a reference in the top left. The two zeta chains are labeled ZA - ζ (εδ)- and ZB - ζ (εγ)-. AlFoM models are shown for TCR in association with PITCR (rank 2) and the negative control peptides PITCRG41P (rank 2), pHLIP (rank 1), and TYPE7 (rank 1), all in red. The extracellular domains are shown semi-transparently. The leucine zipper appended to the ζ chains to constrain dimer formation (see methods) is not shown in A or B.

The online version of this article includes the following source data and figure supplement(s) for figure 9:

**Figure supplement 1.** Comparison between cryoEM structure and AlphaFold-Multimer prediction of T cell receptor (TCR).

**Source data 1.** Interactions and CD3 ζ chain displacements predicted by AlphaFold-Multimer.

**Figure supplement 2.** Comparison of AlphaFold-Multimer model of T cell receptor (TCR) associated with peptide inhibitor of TCR (PITCR) or PITCRG41P.

**Figure supplement 3.** AlphaFold-Multimer IDDT predicted values for all the models generated.

in the presence of DDM to evaluate the specificity and efficacy of the use of the anti-CD3ε antibody for our immunoprecipitation assay. We observed the targeted bands in the immunoblot results for anti-CD3ε-IP, and no bands in the negative control IgG-IP (*Figure 8A*). These results suggest that our protocol successfully IPs the TCR complex. Once we validated our method, we tested the effect of OKT3 stimulation and PITCR treatment. We observed that OKT3 increased the levels of CD3 ζ when compared to both TCRβ2 and CD3ε ( ζ /β2 and ζ /ε) (*Figure 8*). These results suggested a more robust ζ ζ interaction with the rest of complex in response to OKT3 stimulation. Based on this result, we reasoned that PITCR could act by reversing the changes in quaternary robustness that occur in the membrane region upon TCR activation. In agreement with our hypothesis, we observed that the ζ /β2 ratio was decreased when cells were incubated with PITCR and OKT3. However, ζ /ε did not change (*Figure 8*). Taken together, these results suggested that PITCR disrupts the allosteric changes in TM quaternary robustness that occur upon TCR activation.

How does PITCR bind to and inactivate TCR? To address this question, we employed AlphaFold-Multimer (AlFoM) (*Evans et al., 2022*) to predict the binding site of PITCR in the TCR complex. We first assessed if this artificial intelligence program generated robust predictions of the TCR. When we used AlFoM to predict the de novo structure of TCR, we found that it generated a structural model of the TCR that agrees closely (RMSD <1.3 Å) with the cryoEM structure (*Dong et al., 2019*; *Figure 9—figure supplement 1*), after the system was modified, as detailed in the Materials and methods section, to incorporate an experimental constraint, that is, homodimerization of the ζ chains.

Once we optimized AlFoM for TCR, we generated an AlFoM prediction that included PITCR. In the model, PITCR binds to the TM region of TCR, where it interacts tightly with the two ζ chains (*Figure 9*). Interestingly, AlFoM predicted that PITCR induced a conformational change in both ζ chains (*Figure 9B*). Specifically, the CD3ζ that is closer to CD3εγ – z (εγ) – underwent a maximum displacement of ~8 Å, while ζ (εδ) was displaced a maximum of ~6 Å (*Figure 9—source data 1*). To evaluate if this effect was specific, we repeated AlFoM in the presence of the inactive PITCRG41P variant. While the control peptide was predicted to bind to a similar site in TCR, it adopted a different orientation and it did not cause the displacement of the ζ chains observed for PITCR (*Figure 9B* and *Figure 9—source data 1*). Furthermore, several PITCR residues within ~3 Å of a CD3ζ chain were not predicted to interact in the case of PITCRG41P (*Figure 9—source data 1*). When we applied AlFoM to a second negative control peptide, pHLIP (*Figures 3–4*), we observed similar results to PITCRG41P. pHLIP docked into a similar site in TCR, where it interacted weakly with the ζ chains without significantly affecting their position in TCR (*Figure 9B*). To further evaluate if the ζ chain conformational change was specific to PITCR, we also performed AlFoM with the TYPE7 peptide. TYPE7 is a pH-responsive TM peptide that it is not expected to interact with TCR, since it specifically binds to the TM region of the human EphA2 receptor, causing activation of this receptor tyrosine kinase (*Alves et al., 2018*). AlFoM predicted that TYPE7 localizes to a different face of the TCR TM helical bundle and causes minimal displacement of the zeta chains (*Figure 9B*). Taken together, our data present a plausible scenario for TCR interaction and inactivation whereby PITCR binds to the TM helices of the ζ subunits, causing a specific conformational change in the CD3ζ chains (*Figure 9—figure supplement 2*).

## Discussion

For this study we developed a novel conditional TM peptide to target the human TCR. The design of PITCR involved strategically introducing glutamic acid residues into the TM sequence of the human CD3ζ chain, as previously described for TYPE7 (*Alves et al., 2018*). We additionally introduced small and polar residues (*Figure 1—figure supplement 1A*). PITCR selectively inserted into synthetic lipid vesicles at acidic pH (*Figure 1—figure supplement 1*). However, we observed that TCR activation in Jurkat cells was severely disrupted by acidic pH (data not shown). We therefore performed experiments at physiological pH, where we observed that PITCR efficiently targeted cells (*Figure 5* and *Figure 6*). This observation is not unexpected. The acidity-responsive peptide TYPE7 also targeted cellular membranes at neutral pH because the presence in the membrane of its target receptor shifts the pH responsiveness to cause membrane insertion at pH 7.4. We suggest that a similar situation might occur for TCR. However, we speculate that PITCR will more effectively inhibit T cells that reside and survive in acidic environments. The solubility of PITCR is a useful property to facilitate delivery to diseased tissues. Furthermore, the pH responsiveness of PITCR could potentially be used for targeted therapies in pathologies that are characterized by acidic extracellular environments, including inflammatory (*Andreev et al., 2007*) and autoimmune diseases (*Marunaka, 2015*), and solid tumors (*Cheng et al., 2015*). PITCR could also be used to overcome a critical limitation of allogeneic CAR T cell therapy, since TCR inhibition is required to prevent graft-versus-host disease as a side effects of this therapy. The use of PITCR would additionally overcome the risk resulting from genetic manipulation (i.e., by viral gene transfer) of allogeneic T cells before injection into patients (*Michaux et al., 2022*).

PITCR caused robust inhibition throughout the signaling cascade that is triggered when TCR is activated, from phosphorylation of the ζ chain to calcium influx. Upon TCR ligation, immunoblot analysis showed that PITCR inhibited phosphorylation at multiple sites in Zap70, LAT, SLP76, and PLCγ1 (*Figure 2*), but not of Lck (*Figure 2—figure supplement 2*). The observed specific inhibition of TCR downstream signaling suggests that PITCR is unlikely to inhibit/activate a broad range of kinases or phosphatases. Rather, PITCR is likely to specifically inhibit TCR triggering, while maintaining Lck association with coreceptors and its phosphorylation (*Ashouri et al., 2022*; *Guy et al., 2022*).

The results of the immunoprecipitation experiments in the presence of DDM (*Figure 8*) showed that upon TCR activation with OKT3, ζζ strengthened its association with the β subunit. We observed that PITCR caused the opposite effect. This result shows that PITCR acts by reversing the allosteric changes in TM compactness induced by OKT3 activation. However, there was an interesting difference in the interaction with the ε subunit, which increased with OKT3 activation but was not altered by PITCR. This observation suggests that not all the TM interactions are equally important with regard

to contributing to TCR activation by OKT3. Moreover, our results suggest that the interface between the $\zeta$ and β subunits is a target for pharmacological inhibition of the TCR.

We based the DDM immunoprecipitation assay on a previous report (*Lanz et al., 2021*), but our protocol contains significant differences. We performed the immunoprecipitation for the Jurkat TCR, while the previous protocol involved IP of an HA-tagged TCR, which was also activated by different means to ours. Probably as a result of these differences, our DDM immunoprecipitation result showed an opposite effect into how TCR activation affected the association of TCRαβ with $\zeta\zeta$ (*Brazin et al., 2018*; *Lanz et al., 2021*). Nevertheless, the two experimental lines of evidence still agree in showing changes in the quaternary stability of the TCR upon activation, and support that this effect could be an important element of the allosteric activation of the TCR. Overall, our data indicate that binding of PITCR to the TM region of TCR results in different allosteric changes to those that occur upon TCR activation, and we propose this effect reduces signal transduction into the intracellular region. However, caution must be exercised when interpreting experiments when TCR is activated with the anti-CD3 antibody OKT3, since we cannot rule out that differences might exist in the allosteric changes caused by activation with OKT3 or pMHC.

Even though the activation of TCR is an intricate process (*Chai, 2020*; *Dong et al., 2019*; *Mariuzza et al., 2020*; *Reinherz, 2019*), significant progress has been made in understanding the conformational changes it entails. For example, in response to TCR engagement, the juxtamembrane domains of the $\zeta\zeta$ homodimer have been experimentally reported to switch from a divaricated to a parallel conformation (*Lee et al., 2015*). TCR activation additionally involves a conformational change of the ITAMs of CD3ε that releases the interaction of basic residues in the intracellular domain of CD3ζ (*Zhang et al., 2011*) and CD3ε (*Xu et al., 2008*) giving access to Lck for phosphorylation. The AlFoM model identifies the $\zeta$ chain TM domains as the binding site of PITCR, and it is plausible that this interaction may hinder the activating conformational change in $\zeta\zeta$. This hypothesis might further explain the observed decrease of $\zeta$/β2 in the PITCR-treated Jurkat cells upon TCR activation.

The human adaptive immune system contains numerous types of T cells. T cells present a broad repertoire of TCR (*Davis and Bjorkman, 1988*). For example, in human peripheral blood samples, $10^4$ varieties of TCRβs can be found (*Kidman et al., 2020*), while more than $10^{15}$ combinations of TCRαβ could theoretically be formed (*Davis and Bjorkman, 1988*; *Wong et al., 2007*; *Freeman et al., 2009*). The TCR repertoire plays a critical role in the adaptive immune response, but it also brings challenges to achieve immunosuppression to treat inflammatory and autoimmune diseases. Our data show that PITCR interacts with different types of TCRs: it inhibited TCR signaling in Jurkat-wild-type (WT) (*Figure 1*, *Figure 2*, and *Figure 3*), and co-localized with the cSMAC structure formed by TCR(AND) in primary CD4$^+$ T cells (*Figure 6*). Furthermore, PITCR reduced CD69 upregulation in OT1-TCR Jurkat cells (*Figure 4*). These results suggest that PITCR may possess the ability to interact with a broad range of TCR types. We hypothesized that this would be possible because the PITCR design is based on targeting the TM region of the TCR, a sequence that is largely conserved. However, it is important to point out that since we employed different types of T cells for our experiments, our data does not rule out that differences exist in the effect of PITCR on different subtypes of TCR.

Taken together, we report the rational design of a membrane ligand that inhibits TCR activation. PITCR has potential clinical value to treat autoimmune and inflammatory diseases or to avoid transplant rejection. The strategy used to design PITCR can be applied to develop targeted ligands for other receptors causative of disease.

## Materials and methods
### Cell lines

Human male leukemic Jurkat T cells (Clone E6-1, TIB-152) were obtained from the American Type Culture Collection (Manassas, VA, USA). Jurkat.OT1-TCRα-GFP-TCRβ.hCD8$^+$ (J.OT1.hCD8$^+$) cells and T2-K$^b$ cells were kindly provided by Arthur Weiss (UCSF). Cell lines were maintained in RPMI 1640 (Gibco 11875119) supplemented with 10% fetal bovine serum (Gibco 10437028), 1% L-glutamine (Gibco 25030081), and 1% penicillin and streptomycin (Gibco 15140122) in a 37°C and 5% CO$_2$ humid tissue culture incubator (Panasonic Healthcare, Wood Dale, IL, USA). The identity of Jurkat cells was authenticated using ATCC services. Mycoplasma contamination was ruled out by PCR (Abcam 289834).

### Peptide synthesis

Peptides were synthesized by Thermo Fisher Scientific (Waltham, MA, USA) and were confirmed by matrix-assisted laser desorption ionization-time-of-flight (MALDI-TOF) mass spectrometry and reverse-phase high-performance liquid chromatography (HPLC). Purities of the peptides are over 95%. PITCR sequence: DPKLSYLLDGILFGYGVELTALFLEVGFSESAD.

### Intracellular calcium assay

Jurkat-WT cells were washed twice with PBS and then stained with the calcium sensor dye Indo-1 AM (Invitrogen I1223), at 37°C and 5% $CO_2$ for 30 min as described (*Lo et al., 2018*; *Lo et al., 2019*). The final concentration of Indo-1 AM was 1 µM. Stained Jurkat cells were washed twice with PBS and then were transferred to a 96-well flat-bottom black plate. PITCR was added and incubated at 37°C and 5% $CO_2$ for 20 min. Next, the plate was transferred to a prewarmed (37°C) and 5% $CO_2$ Cytation V plate reader (BioTeK, Winooski, VT, USA) and incubated for another 10 min. The final concentration of PITCR in the each well was 10 µM. The baseline for each well was recorded for the first 100 s, followed by auto-injection of anti-CD3 (OKT3 clone, Tonbo-70-0037). The final concentration of anti-CD3 was 1 µM. Ionomycin (Invitrogen I24222) was used as a positive control to prove that Jurkat cells respond to calcium influx. The fluorescent signal collected from Jurkat-WT cells without staining was subtracted from the signal collected from Indo-1 AM stained cells since Jurkat-WT cells have auto-fluorescent signals.

### SDS-PAGE immunoblot analysis of proximal signal molecules of TCR-CD3, ζ-Y83, and ζ-Y142

Jurkat-WT cells were washed twice with PBS and treated with PITCR at 37°C and 5% $CO_2$ for 30 min, followed by stimulation with anti-CD3 antibody (OKT3 clone) for 5 min. The final cell density was 5×10^6 cells/mL and the final concentration of PITCR was 10 µM. The final concentration of anti-CD3 was 1 µM. Cells were lysed in 1 % NP-40 lysis buffer (50 mM Tris-Cl pH 7.4, 150 mM sodium chloride, 2 mM PMSF, 5 mM EDTA, 0.25% sodium deoxycholate with proteinase inhibitors [Thermo Scientific A32955] and phosphatase inhibitors [Sigma-Aldrich P0044]) for 30 min on ice, followed by centrifugation of 16.2×10^3 × g, 30 min, 4°C. Supernatants were collected and detergent-compatible protein assay (Bio-Rad5000112) was performed to quantify the protein concentration of each sample. Equal amounts of protein samples were run in 10%, 12%, or 15% SDS-PAGE gels and transferred to 0.45 µm nitrocellulose membranes. Membranes were blocked with 3% bovine serum albumin (BSA) dissolved in TBS, followed by overnight incubation with primary antibodies at 4°C. IRDye 800CW or IRDye 680LT secondary antibodies were used to incubate the blots at the second day, followed by detection with an Odyssey Infrared Scanner (Li-Cor Biosciences, Lincoln, NE, USA). All primary antibodies were diluted in 5% BSA dissolved in 0.1% TBST except specific mentions and all secondary antibodies were diluted in 5% non-fat milk dissolved in 0.1% TBST unless mentioned otherwise.

### Co-localization assay and analysis

All steps were performed at room temperature (RT), unless noted otherwise. Jurkat-WT cells were washed once with PBS, resuspended in RMPI1640 phenol-red free media, and treated with dylight680 labeled PITCR (PITCR680) at 37°C and 5% $CO_2$ for 15 min, followed by stimulation in presence or absence of anti-CD3 (OKT3 clone) for 5 min. The final concentration of anti-CD3 was 1 µM. The final cell density was 5×10^6 cells/mL and the final concentration of PITCR680 was 10 µM. PITCR680 treated cells were washed twice with cold PBS and resuspended in cold RPMI1640 phenol-red free media. 100 µL of cells were transferred to each chamber of µ-Slide 8 Well ibiTreat (ibidi80826) and rested for 10 min. Next, RPMI1640 phenol-red free media was removed gently. 150 µL cold fixing buffer (1.998% - formaldehyde and 0.2% - glutaraldehyde in filtered PBS) was added and incubated for 10 min. Fixed PITCR680 treated Jurkat cells were washed twice with cold DPBS/Modified (HycloneSH30264.01). 150 µL permeabilization buffer (0.1% Triton X100 in PBS) was incubated with the sample for another 15 min. Each chamber was washed with blocking buffer (5% goat serum in 0.01% PBST) twice. The sample was blocked 1 hr, followed by washing once with antibody dilution buffer (1% BSA in 0.01% PBST). The samples were incubated with 1:200 diluted anti-CD3ε (UCHT1 clone, sc-1179) primary antibody in the wet box at 4°C, overnight. On the second day, each chamber was washed twice with cold DPBS/Modified (HycloneSH30264.01). 1:200 diluted secondary antibody (Invitrogen A32723)

was incubated with samples in a foil covered wet box for 1 hr, followed by washing twice with cold DPBS/Modified (HycloneSH30264.01). 1:1000 diluted DAPI (Thermo Scientific 62248) was stained with samples for 2 min, followed by washing once with cold DPBS/Modified. The samples were mounted with Vectashield (H-1000), followed by sealing the chambers with parafilm until imaging.

Samples were imaged using a Leica SP8 White Light Laser Confocal Microscope (Leica, Wetzlar, Germany) equipped with a 63× oil immersion objective, zoom 5 through LAS X software. Z-stack scanning was applied, followed by a lightning deconvolution analysis. Each image was chosen from one time point at each Z-stack section. Graphic profile curves of the Region of Interest in Zoom-in images were analyzed using Image J RGB Profile Plot Plugin. The Mander's M1 and Pearson's $r$ coefficients were calculated using Image J Just Another Colocalization Plugin (JACoP).

## CD69 activation flow cytometry assay

$2 \times 10^6$ cells/mL T2-K$^b$ cells were washed twice with PBS and treated with a series of diluted OVA derived peptide (SIINFEKL) at 37°C and 5% $CO_2$ for 1 hr. For this assay we used J.OT1.hCD8$^+$ cells, which are engineered human Jurkat T cells, as described (*Lo et al., 2018*). $5 \times 10^6$ cells/mL J.OT1. hCD8$^+$ cells were washed twice with PBS and treated with PITCR at 37°C and 5% $CO_2$ for 30 min. PITCR-treated J.OT1.hCD8$^+$ cells were added to OVA-treated T2-K$^b$ cells and the ratio of these two cells was 1:1. Final concentration of PITCR was 10 μM. The incubation time was 3 hr. Next, anti-CD69 - Allophycocyanin (Biolegend 310910) and Isotype - Allophycocyanin (Biolegend 400122) were applied to stain the cells, respectively followed by LSRII Flow Cytometer (BD Bioscience, Franklin Lakes, NJ, USA) analysis. Data was quantified using FlowJo_v10.8.0 software.

## Co-IP of TCR-CD3 complex and immunoblot analysis

Jurkat-WT T cells were washed twice with PBS and treated with Dylight680 labeled PITCR at 37°C and 5% $CO_2$ for 30 min, followed by stimulation with 1 μM anti-CD3 antibody (OKT3 clone). Cells were lysed in 2.5% DIBMA (AnatraceBMA101) lysis buffer 20 mM Tris-Cl pH 8.0, 137 mM NaCl, 2 mM EDTA, 1 mM PMSF, 5 mM iodoacetamide, 1 mM NaF, proteinase inhibitors (Promega G6521) and phosphatase inhibitors (Sigma-Aldrich P0044 and P5726) at 37°C for 2 hr followed by rotating at least 12 hr in the cold room (4°C). Lysates were ultracentrifuged $10,000 \times g$, 4°C, 45 min to get rid of debris, and the supernatants were collected and ultracentrifuged again $10,000 \times g$, 4°C, 1 hr. The protein concentrations were quantified using a detergent compatible protein assay. 40 μL whole lysates were saved to detect the TCR-CD3 complex. 1% BSA blocked protein G agarose (Thermo Scientific 20398) and anti-CD3ε (UCHT1 clone, sc-1179) were added to the rest of lysates. Samples were gently rotated at cold room (4°C) for 16 hr. Samples were centrifuged $5000 \times g$, 4°C, 3 min, followed by washing twice with cold wash buffer (20 mM Tris-Cl pH 8.0, 137 mM NaCl, 2 mM EDTA, 1 mM PMSF, 5 mM iodoacetamide, 1 mM NaF) and rinsing once with cold wash buffer. SDS sample buffer was applied to elute the captured protein followed by incubation at 95°C for 5 min.

Equal amounts of whole lysate samples and captured protein samples were loaded in 15% SDS-PAGE glycine gels and transferred to 0.45 μm or 0.2 μm nitrocellulose membranes. 3% BSA was used to block the membranes, followed by incubation with primary antibodies: anti-TCRβ, anti-CD3ε and anti-CD3 ζ (6B10.2 clone) overnight at 4°C. In the whole lysates' group, the housekeeping protein β-actin was also probed. PITCR was directly detected using channel 700 nm of an Odyssey Infrared Scanner (Li-Cor Biosciences, Lincoln, NE, USA). The second day, same approaches were performed as described in SDS-PAGE immunoblot analysis of proximal signal molecules of TCR-CD3, ζ -Y83, and ζ -Y142.

## Liposome preparation

1-Palmitoyl-2-oleolyl-glycero-3-phosphocholine (POPC) and 1-palmitoyl-2-oleoyl-*sn*-glycero-3-phospho-L-serine (POPS) were purchased from Avanti Polar Lipids, Alabaster, AL, USA. POPC and POPS were dissolved in cold chloroform and stocks prepared at 32.89 mM. POPS and POPC were mixed in a round-bottom test tube, dried together under a stream of argon gas, and placed in a vacuum overnight. The dried lipid film was rehydrated in 10 mM sodium phosphate buffer (pH 7.4), followed by extrusion with a Mini-Extruder (Avanti Polar Lipids, Alabaster, AL, USA) through a 0.1 μm Nuclepore Track-Etch membrane (Whatman, Maidstone, UK). The final large unilamellar vesicles (LUVs) stock concentration was 4 mM. LUVs contain 10% POPS and 90% POPC.

## Circular dichroism

Circular dichroism (CD) experiments were performed as described previously (*Nguyen et al., 2015*). Briefly, the secondary structure of PITCR in aqueous solution (10 mM sodium phosphate, pH 7.4) and liposomes (10% POPS and 90% POPC LUVs at pH 5.0 and pH 7.4, respectively) was determined in a Jasco J-815 spectropolarimeter at RT. The CD spectra were measured from 195 nm to 260 nm in a 2 mm path length quartz cuvette. The peptide to lipid molar ratio was 1: 200 and the final concentration of the peptide was 5 µM. To obtain the desired pH, the experimental samples were adjusted by adding either 100 mM sodium phosphate (pH 7.4) or 100 mM sodium acetate (pH 4.0). The appropriate liposome or buffer backgrounds were subtracted. Molar ellipticity was calculated with the following equation: $[\theta] = \theta/\left[10lc\left(N-1\right)\right]$, where $\theta$ is the measured ellipticity in millidegree, $l$ is the cell path length, $c$ is the protein concentration, and $N$ is the number of amino acids ($N$=33).

## pK$_{CD}$ determination assay

The apparent pK$_{CD}$ is defined as a pH midpoint, where half of the peptide changes its secondary structure in presence of liposomes (*Scott et al., 2017*). The liposome preparation (10% POPS and 90% POPC LUVs) was followed as described in liposome preparation. To reach a series of different desired pH values, the experimental samples were adjusted with either 100 mM sodium phosphate or 100 mM sodium acetate. The final pH was measured by a 2.5 mm bulb pH electrode (Microelectrodes, Bedford, NH, USA) after recording CD spectrum. The CD spectra were recorded from 220 nm to 262 nm. The appropriate liposome blanks were subtracted. The ellipticity values at 222 nm were subtracted from that at 262 nm. The subtracted ellipticity were plotted for a range of pH values. The pK$_{CD}$ was fitted in the following equation: $Signal = \frac{[(F_a+S_a \times pH)+(F_b+S_b \times pH) \times 10^{[m \times (pH-pK)]}]}{[1+10^{[m \times (pH-pK)]}]}$, where $F_a$ is the acidic baseline, $F_b$ is the basic baseline, $m$ is the slope of the transition, and $pK$ is the midpoint of the curve.

## Peptide conjugation

Cysteine was added to the N-terminal of PITCR, termed NEC-PITCR (sequence: ECDPKLSYLLDG ILFGYGVELTALFLEVGFSESAD). NEC-PITCR was labeled with dylight680 maleimide (Thermo Scientific-46618) and AZDye 555 maleimide (Fluoroprobes-1168-1). Both dyes labeled peptides were purified using reverse phase HPLC purification to remove unconjugated dye. The molecular weight was confirmed by MALDI-TOF. After that, samples were aliquoted, lyophilized, and stored at –80°C.

## MALDI-TOF mass spectrometry

PITCR-associated peptides were dissolved in 1 mM sodium phosphate buffer (pH 7.4, filtered). The matrix α-cyano-4-hydroxycinnamic acid (TCI C1768) was dissolved in 75% HPLC-level acetonitrile coupled with 0.1% TFA and sonicated 15 min, RT. The dissolved samples were mixed with the dissolved matrix. After that, the mixed matrix samples were loaded onto the MSP target plate (Bruker, Billerica, MA, USA) drop by drop and dried using filtered air. The Bruker Microflex MALDI-TOF mass spectrometer (Bruker, Billerica, MA, USA) was calibrated with ProteoMass MALDI-MS calibration standards (Sigma-Aldrich I6279-5X1VL, I6154-5X1VL, C8857-5X1VL, and P2613-5X1VL). All PITCR-associated peptides were measured in a negative mode. The pHLIP was measured in a positive mode. Data were analyzed using FlexAnalysis software (Bruker, Billerica, MA, USA) and graphs were plotted using Origin 9.1 (research lab) software.

## HPLC

All peptides were dissolved in 1 mM sodium phosphate buffer (pH 7.4, filtered). Dissolved peptides were injected into an Agilent 1200 series HPLC system (Agilent Technologies, Santa Clara, CA, USA). A semi-preparative Agilent Zorbax 300 SB-C18 column (P.N. 880995-202) was used to purify the PITCR-associated peptides. A stable-bond analytical Agilent Zorbax 300 SB-C18 column (P.N. 880995-902) was used to identify the purity of each peptide. The running procedure used a gradient (solvent A: 0.05% TFA HPLC-level water plus solvent B: 0.05% TFA HPLC-level acetonitrile) from 5% B to 100% B. PITCR-associated peptides were eluted around 80% B.

## Immunoprecipitation and immunoblot analysis

Jurkat-WT T cells were washed twice with PBS and treated with PITCR at 37°C and 5% $CO_2$ for 30 min, followed by stimulation with 1 µM anti-CD3 antibody (OKT3 clone). Cells were lysed in cold 0.5% dodecyl-β-D-maltopyranoside (DDM, VWR-97063-172) lysis buffer (20 mM Tris-Cl pH 8.0, 10 mM NaF, 166.67 mM NaCl, 20 mM iodoacetamide, Benzonase endonuclease 50 U/mL [Sigma-1016970001]), proteinase inhibitors (Thermo Scientific A32955), and phosphatase inhibitors (Sigma-Aldrich P0044 and P5726) for 30 min on ice, followed by centrifugation of $16.2 \times 10^3 \times g$, 15 min, 4°C. Supernatants were collected and detergent compatible protein assay was performed to quantify the protein concentration of each sample. 30–50 µL whole lysates were saved to detect the TCR complex. Protein G agarose and anti-CD3ε (UCHT1 clone) were used as described in the methodology of Co-IP of TCR. Samples were continuously rotated at 4°C for 4 hr. Other steps follow the methodology of Co-IP of TCR, except the wash buffer (20 mM Tris-Cl pH 8.0, 10 mM NaF, 166.67 mM NaCl, 20 mM iodoacetamide, Benzonase endonuclease 50 U/mL, Sigma-1016970001) and the elution condition (70°C for 10 min). Immunoblot analysis was performed as described in the methodology of Co-IP of TCR.

## Antibodies

| Antibodies | Sources | Catalogue # | Dilutions |
|---|---|---|---|
| Zap70 (pY319) | Cell Signaling Technology | 2717 | 1:1000 (WB) |
| Zap70 (pY493) | Cell Signaling Technology | 2704 | 1:1000 (WB) |
| Zap70 (total) | Cell Signaling Technology | 3165 | 1:1000 (WB) |
| LAT (pY191) | Cell Signaling Technology | 3584 (discontinued) | 1:1000 (WB) |
| LAT (pY132) | Abcam | ab4476 | 1:2000 (WB) |
| LAT total | Cell Signaling Technology | 45533 | 1:1000 (WB) |
| Lck (pY394) | R&D systems | 755103 | 1:2000 (WB) |
| Lck (pY505) | Cell Signaling Technology | 2751 | 1:1000 (WB) |
| Lck (total) | Cell Signaling Technology | 2984 | 1:1000 (WB) |
| β-Actin | Abcam | ab6276 | 1:5000 (WB) |
| CD3ε (OKT3) | Tonbo Biosciences | 70-0037 | 1:50 (stimulation) |
| CD3ε (UCHT1) | Santa Cruz Biotechnology | sc-1179 | 1: 200 (IF), IP (see methodology) |
| CD3ε (CD3-12) | Cell Signaling Technology | 4443 | 1:1000 (WB) |
| ζ (6B10.2) | Santa Cruz Biotechnology | sc-1239 | 1:500 (WB) |
| ζ (pY142) | BD Biosciences | 558402 | 1:1000 (WB) |
| ζ (pY83) | Abcam | ab68236 | 1:1000 (WB) |
| TCR β | Cell Signaling Technology | 77046 | 1:2000 (WB) |
| SLP76 (pY128) | BD Biosciences | 558367 | 1:2000 (WB) |
| PLCγ1 (pY783) | Cell Signaling Technology | 2821 | 1:1000 (WB) |
| IgG | Cell Signaling Technology | 5415 | IP (see methodology) |
| CD69-APC (FN50) | Biolegend | 50-166-584 | 1:100 (Flowcytometry) |
| IgG-APC | Biolegend | 50-168-838 | Flowcytometry |
| Goat anti-Mouse IgG (H+L) Highly Cross-Adsorbed Secondary Antibody, Alexa Fluor Plus 488 | ThermoFisher Scientific (Invitrogen) | A32723 | 1: 200 (IF) |
| IRDye 800CW Goat anti-Rabbit IgG Secondary Antibody | LI-COR Bioscience | 926-32211 | 1:5000 (WB) |

*Continued on next page*

*Continued*

| Antibodies | Sources | Catalogue # | Dilutions |
| --- | --- | --- | --- |
| IRDye 800CW Goat anti-Mouse IgG Secondary Antibody | LI-COR Bioscience | 926-32210 | 1:5000 (WB) |
| IRDye 800CW Goat anti-Rat IgG Secondary Antibody | LI-COR Bioscience | 925-32219 | 1:5000 (WB) |

## TIRF imaging of AND-TCR T cells

AND-TCR T cells were incubated with PITCR-AZDye555 by mixing 50 µL of a 100 µM solution in 10 mM sodium phosphate buffer pH 7.4 with 450 µL of 2.5 M cells/mL T cells in RVC medium with IL-2 (final concentration of 10 µM PITCR-AZDye555, 2.2 M cells/mL, 37°C, 30 min), rinsed by imaging buffer, then applied to the imaging chamber with SLB functionalized with ICAM-1 (~20 molecules/µm$^2$) and pMHC (19–23 molecules/µm$^2$, MCC peptide 9.1% labeled with Atto647N) at 37°C. The real-time images of just-adhering cells with RICM, TIRF at 561 nm excitation (50 ms exposure), and TIRF at 640 nm excitation (500 ms exposure) channels were recorded at the average frame rate of 1 frame per 4 s. After 15–30 min, the snapshots of the cells forming immune synapses were recorded with the same three channels. The experiment was performed with 5 cells (real-time) and about 50–100 cells (snapshots) from one mouse. The cells with dominant plasma-membrane-bound PITCR signal could be found only in snapshot measurements due to low population. The vehicle control was performed by treating the cells with phosphate buffer instead of PITCR-AZDye555 solution using the cells from the same mouse.

## Reagents

T cell culture medium: DMEM (Gibco, Thermo Fisher) with 10% FBS, 1 mM sodium pyruvate, 2 mM L-glutamine, 1x Corning nonessential amino acids (Fisher Scientific), 1x Corning MEM vitamin solution (Fisher Scientific), 0.67 mM L-arginine, 0.27 mM L-asparagine, 14 µM folic acid, 1x Corning penicillin/streptomycin (100 IU, 0.1 mg/mL respectively) (Fisher Scientific), 50 µM β-mercaptoethanol.

Imaging buffer for TIRF experiment: 20 mM HEPES, 137 mM NaCl, 5 mM KCl, 1 mM MgCl$_2$, 1.8 mM CaCl$_2$, 0.1% wt/vol D-glucose, 0.1% wt/vol BSA, pH 7.4.

## AND-TCR T cell culture

Primary AND-TCR T cells were prepared and cultured basically as previously described (*Smith et al., 2011*). T cells from the lymph nodes and spleens were harvested from (B10.Cg-Tg(TcrAND)53Hed/J) × (B10.BR-H2k2 H2-T18a/SgSnJ) hybrid mice (Jackson Laboratory) and kept in T cell culture medium (day 1). The cells were activated by 2 µM MCC peptide at day 1, then cultured with IL-2 after day 2. Cells were used at day 5 or 6 for imaging. All animal work was performed with prior approval by Lawrence Berkeley National Laboratory Animal Welfare and Research Committee under the approved protocol 177003.

## pMHC and ICAM-1 preparation

ICAM-1 extracellular domain with His10 tag and MHC class II I-Ek with two His6 tags were expressed and purified as previously described (*Nye and Groves, 2008*).

Peptides for pMHC were prepared and loaded to MHC molecule as previously described (*O'Donoghue et al., 2013*) MCC peptide (ANERADLIAYLKQATK) and MCC-GGSC (ANERADLIAYLKQATK-GGSC) were synthesized on campus (D King, Howard Hughes Medical Institute Mass Spectrometry Laboratory at University of California, Berkeley, CA, USA) or commercially (Elim Biopharmaceuticals, Hayward, CA, USA). MCC-GGSC was labeled with Atto647N-maleimide (ATTO-TEC), purified by C18 column reversed-phase HPLC, and identified by MALDI-TOF mass spectrometry.

Excess amount of MCC and MCC-GGSC-Atto647N were separately loaded on MHC molecules in loading buffer (PBS acidified with citric acid to pH 4.5, 1% BSA) overnight at 37°C. Then mixed at a 10:1 molar ratio to achieve 9.1% labeling efficiency. The mixture was diluted-concentrated with TBS and 10 kDa MWCO filters (Spin-X UF, Corning, NY) for two times to remove excess peptides, then used for bilayer functionalization.

## Supported lipid bilayer preparation

Small unilamellar vesicles with 98 mol% 1,2-dioleoyl-*sn*-glycero-3-phosphocholine (Avanti Polar Lipids) and 2 mol% 1,2-dioleoyl-*sn*-glycero-3-[(*N*-(5-amino-1-carboxypentyl)iminodiacetic acid)succinyl] nickel salt (Avanti Polar Lipids) were prepared by sonicating a 0.5 mg/mL lipid suspension in water followed by centrifugation (21,000 × *g*, 20 min, 4°C). Then, supported lipid bilayer (SLB) was prepared upon 25 mm #1.5 glass coverslip set into Attofluor cell chamber (Invitrogen, Thermo Fisher). Coverslips were cleaned by sonication in 1:1 water:2-propanol then etched with piranha solution (1:3 mixture of 30% $H_2O_2$ and sulfuric acid), rinsed by water and set into clean chambers. SLB was formed by adding 1:1 mixture of SUV solution and TBS into chambers and incubating for more than 30 min. SLB were rinsed with TBS then incubated with 10 mM $NiCl_2$ in TBS for 5 min. Chambers were then incubated in imaging buffer for more than 30 min for blocking defects by BSA, then used for functionalization. ICAM-1 (~10 nM) and pMHC (concentration adjusted by determined densities) were added to chambers and incubated for 30 min, then rinsed by imaging buffer. The ICAM-1 density is estimated to be ~20 molecules/$\mu m^2$ based on a previously reported estimate (*Lin et al., 2019*). pMHC labeled by Atto647N was imaged by TIRF to determine the molecular density. The density was determined by extrapolating the calibration curve of density-intensity relationship from lower densities where the molecular density can be directly determined by single molecule localization (below 0.5 molecules/$\mu m^2$) using TrackMate (*Tinevez et al., 2017*).

## TIRF microscopy and image processing

TIRF microscopy was performed on a motorized inverted microscope (Nikon Eclipse Ti-E; Technical Instruments, Burlingame, CA, USA) with Lumen Dynamics X-Cite 120LED Fluorescence Illumination System (Excelitas Technologies, Waltham, MA, USA) and a motorized stage (MS-2000; Applied Scientific Instrumentation, Eugene, OR, USA). A laser launch with 561 nm and 640 nm diode lasers (Coherent OBIS, Santa Clara, CA, USA) was aligned into a custom-built fiber launch (Solamere Technology Group Inc, Salt Lake City, UT, USA). For TIRF imaging, laser excitation was illuminated through a four-band beam splitter (ZT405/488/561/640rpc) to the objective lens (NA 1.49, 100×, oil immersion, Apochromat TIRF, Nikon), then filtered through an emission filter (ET600/50M or ET700/75M). For RICM, LED excitation was illuminated through D546/10× excitation filter and 50/50 beam splitter. Emission was captured on an EM-CCD (iXon Ultra 897; Andor Inc, South Windsor, CT, USA). All optical filters were purchased from Chroma Technology Corp (Bellows Falls, VT, USA). The sample and objective lens were kept at 37°C with temperature controller system (CU-109, Live Cell Instrument, Republic of Korea). The equipment was controlled using the software MicroManager (*Edelstein et al., 2010*). Laser power and exposure time was set to 2 mW, 50 ms for 561 nm excitation and 1 mW, 500 ms for 640 nm excitation. The pixel size was 0.16 μm square and the field of view was 81.92 μm square (512×512 pixels).

Cell footprint was determined from RICM images with the following procedures. RICM image was gaussian-blurred (sigma: 2 pixels), manually background-subtracted, and converted to absolute values pixel-wise. The obtained intensity images were segmented by semi-automatic way: the inner region and the encompassing region of the cell of interest were manually selected. Image was thresholded by the intensity in the inner region multiplied by an arbitrary factor of 0.5. The obtained segment was filtered within the encompassing region, then cleaned by binary opening (kernel: 3×3 pixels square). The regions smaller than 200 pixels were deleted and the remaining regions (multiple regions were allowed to exist if any) were used as the cell footprint.

The background signal was measured using the chamber containing only imaging buffer and subtracted from TIRF images. The inhomogeneity of the TIRF illumination were corrected using the images from the solution of rhodamine B (561 nm excitation, from Sigma-Aldrich) and 3,3'-diethylthiadicarbocyanine iodide (640 nm excitation, from Sigma-Aldrich).

## NFAT activation assay

To assay activation of AND-TCR primary murine CD4[+] T cells, cells were transduced with a LAT-eGFP-P2A-NFAT-mCherry bicistronic construct on day 3 of primary cell culture as previously described (*Smith et al., 2011*). Cells were assayed on day 5. All animal work was performed with prior approval by Lawrence Berkeley National Laboratory Animal Welfare and Research Committee under the approved protocol 177003.

For each imaging chamber, 2.5 million cells were resuspended to 5 million/mL in 450 µL imaging buffer and 50 µL of 100 µM unlabeled PITCR in 10 mM sodium phosphate buffer pH 7.4. Control samples were treated the same, with PITCR omitted. Cells were incubated for 30 min at 37°C and then directly added to SLBs functionalized with ICAM and pMHC in Attofluor chambers containing 500 µL imaging buffer equilibrated to 37°C. Cells interacted with the bilayer for 20 min before acquiring snapshots to analyze for NFAT activation state. Cells transduced with reporter proteins were identified using the LAT signal in the 488 TIRF channel to minimize bias in which cells were imaged and subsequently analyzed. Single sets of RICM, 488 TIRF, and 561 epifluorescence images were taken for at least 30 fields of view and at least 50 live cells 20–50 min after adding cells to the SLB. Three z-positions were acquired for 561 epifluorescence at 0, 3, and 6 µm above the TIRF plane in order to clearly resolve each cell's nucleus and the distribution of NFAT-mCherry between the cytoplasm and nucleus. Only cells with substantial contact with the bilayer, as defined by the RICM footprint, were included in the analysis of the fraction of activated cells. Cells were defined as active if the NFAT-mCherry signal in the nucleus was equal to or greater than the signal in the cytoplasm, assessed manually, indicating that the NFAT-mCherry reporter protein was translocating to the nucleus. The fraction of activated cells was determined for each bilayer density and pre-treatment condition and error bars denote the standard error of the mean. The density of pMHC on each bilayer was determined before the addition of cells. Snapshots of pMHC in at least 20 fields of view were taken in the 640 TIRF channel at 20 mW power at the source and 20 ms exposure time. Particles were counted using TrackMate and density was determined as the particle count divided by the total area of all fields of view. A calibration curve relating density and intensity was used to measure the density of high-density bilayers for which single particles are not able to be resolved ($\sim 0.7\ \mu m^{-2}$).

## AlphaFold2-Multimer predictions

Structures of the TCR with and without addition of peptide were predicted using AlphaFold2-Multimer. We installed LocalColabFold 1.5.1 (*Mirdita et al., 2022*), which uses AlphaFold v2.3.0, on a Yale Center for Research Computing Linux cluster (*Evans et al., 2022*). The random seed used was 0, and amber relaxation was enabled. Predictions of the TCR were generally consistent with the cryoEM structure of TCR, except for one of the CD3 $\zeta$ chains, which detached from the complex as it lacked the stabilization that results from the native disulfide bond that connects the two $\zeta$ chains. In order to force the CD3 $\zeta$ homodimer to adopt a physiologically relevant conformation, one chain of the *Saccharomyces cerevisiae* basic leucine zipper (bZIP) domain of GCN4 (amino acid sequence RMKQLEDKVEELLSKNYHLENEVARLKKLVGER) was appended immediately C-terminal to each CD3 $\zeta$ TM domain, after residue Arg57 (UniProtKB #P20963). CD3 $\zeta$ residues C-terminal to Arg57 were removed. Polyglycine linkers of two different lengths (four and ten) were used separately to connect the bZIP domains to CD3 $\zeta$ with optimal alignment. Multiple ranks were created for each prediction, and the highest ranking structure where the peptide approaches the side of the TCR TM domains was selected for further analysis. The predicted IDDT scores for all structures are shown on *Figure 9—figure supplement 3*. Predicted structures were visualized using Mol* and distances between peptide side chain atoms and CD3 $\zeta$ side chain or backbone atoms were calculated using the measurements tool (*Sehnal et al., 2021*). For displacement distances, the Mol* measurements tool was used to calculate the maximum distance between all corresponding residues in each model after superimposing the TCRα chains. ChimeraX was used to calculate root-mean-square deviation (RMSD) values (*Pettersen et al., 2021*). The matchmaker command was employed to align pairs of amino acids from corresponding chains and then calculate the RMSD based on the α-carbons. The TCRα chain was used as a consistent reference in RMSD alignments and calculations so that CD3 $\zeta$ chains could be compared between no peptide and plus peptide structures.

## Statistical analysis

All statistical analyses of experiments were performed using GraphPad Prism 9.4.0. p-Values are provided as exact values. 95% confidence level was used to determine statistically significance in all experiments and ns stands for not significant. All statistics correspond to biological replicates only and all *n* values reflect biological replicates. Detailed statistical analyses were illustrated in each result.

## Acknowledgements

This work was supported by NIH grants R35GM140846 (to FNB) and R35CA242462 (to DD), and NIH training grant (T32AI055403) and National Science Foundation Predoctoral Fellowship (DGE-2139841) to PMB, and additionanlly by a Faculty-Graduate student award to YY (University of Tennessee). We thank the Yale Center for Research Computing for guidance and use of the research computing infrastructure. We thank Art Weiss (UCSF) for generous advice to YY and for providing JOT1. CD8 and T2Kb cells, and to L Teyton (Scripps Research) and M Davis (Stanford University) for providing the MHC and ICAM-1 bacmids. We also appreciate the advice provided to YY by Barry Bruce (University of Tennessee), Matthew Call (Walter and Eliza Hall Institute of Medical Research), and Peiqing Sun (Wake Forest Baptist Medical Center). We are also thankful for the technical advice of Tim Sparer and Trevor Hancock (University of Tennessee), Jaydeep Kolape (AMIC, University of Tennessee) and Ed Wright (BRF, University of Tennessee). We appreciate the members of the Barrera lab Jen Schuster, Jordan Pyron, and Boomer Russell for insights on the manuscript.

## Additional information

### Funding

| Funder | Grant reference number | Author |
|---|---|---|
| National Institute of General Medical Sciences | R35GM140846 | Francisco N Barrera |
| National Cancer Institute | R35CA242462 | Daniel DiMaio |
| National Institutes of Health | T32AI055403 | Patrick M Buckley |
| National Science Foundation Predoctoral Fellowship | DGE- 2139841 | Patrick M Buckley |
| University of Tennessee | | Yujie Ye |

The funders had no role in study design, data collection and interpretation, or the decision to submit the work for publication.

### Author contributions

Yujie Ye, Conceptualization, Data curation, Formal analysis, Supervision, Funding acquisition, Investigation, Methodology, Writing – original draft, Writing – review and editing; Shumpei Morita, Conceptualization, Data curation, Investigation, Methodology, Writing – original draft, Writing – review and editing; Justin J Chang, Formal analysis, Supervision, Investigation, Writing – original draft, Writing – review and editing; Patrick M Buckley, Kiera B Wilhelm, Jay T Groves, Conceptualization, Formal analysis, Supervision, Funding acquisition, Investigation, Writing – original draft, Writing – review and editing; Daniel DiMaio, Francisco N Barrera, Conceptualization, Formal analysis, Supervision, Funding acquisition, Investigation, Methodology, Writing – original draft, Writing – review and editing

### Author ORCIDs

Yujie Ye ⬤ http://orcid.org/0000-0002-1067-5867
Shumpei Morita ⬤ http://orcid.org/0000-0003-0070-852X
Patrick M Buckley ⬤ http://orcid.org/0000-0003-2688-5858
Kiera B Wilhelm ⬤ https://orcid.org/0000-0002-8781-7739
Daniel DiMaio ⬤ http://orcid.org/0000-0002-2060-5977
Francisco N Barrera ⬤ http://orcid.org/0000-0002-5200-7891

### Ethics

All animal work was approved by Lawrence Berkeley National Laboratory Animal (Berkeley, CA) Welfare and Research Committee under the approved protocol #177003.

### Decision letter and Author response

Decision letter https://doi.org/10.7554/eLife.82861.sa1
Author response https://doi.org/10.7554/eLife.82861.sa2

## Additional files

### Supplementary files
• MDAR checklist
• Source data 1. This source data contains all original plot data.

### Data availability
All data generated or analysed during this study are included in the manuscript and supporting file; Source Data files have been provided.

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
