## [Editor Report]

The authors combine AlphaFold-Multimer and a previously described technology of designing soluble transmembrane-targeting peptides that interfere with the function of the T cell receptor (TCR). This study provides important insights into the molecular mechanism of T cell activation. The approach is convincing since the designed PITCR peptide has functional effects, in contrast with the predicted negative controls PITCRG41P and pHLIP. The results and the methods from this study will be of interest to those studying the TCR, as well as those seeking to use the TCR or its derivatives in synthetic biology studies and immunotherapy.

---

## [Decision Letter]

**Decision letter after peer review:**

Thank you for submitting your article "Allosteric Inhibition of the T Cell Receptor by a Designed Membrane Ligand" for consideration by *eLife*. Your article has been reviewed by 3 peer reviewers, and the evaluation has been overseen by a Reviewing Editor and Jonathan Cooper as the Senior Editor. The following individual involved in the review of your submission has agreed to reveal their identity: Michael L Dustin (Reviewer #1).

The reviewers agree that the findings are potentially significant but the evidence to support the conclusions is incomplete, as listed in the individual critiques.

There are 4 issues that should be addressed:

1. Different cells were used in different measurements and the results are combined in the model. More experiments may be needed to link the results from the different experiments.

2. Why zeta was selected should be discussed.

3. Anti-CD3 was used for stimulation in most experiments but the results are discussed in the context of pMHC activation. Does anti-CD3 cause the same allosteric changes as pMHC? If this is an issue, the extrapolation to pMHC should be toned down.

4. The hypothesis is that the peptide inhibitor, which resembles the CD3 zeta TM domain and might be expected to displace zeta, actually inhibits allosteric activation. This model really needs structural studies (such as NMR) to confirm. However, we recognize that such studies are likely beyond the scope of the paper, or what could be achieved in a 2-3 month period. The paper would be strengthened if additional data or information could be provided regarding the mechanism of interaction between the peptide and the TCR.

*Reviewer #1 (Recommendations for the authors):*

The study has some sophisticated elements to address an important question. Limitations include a lack of explanation of the process to arrive at this candidate, the relatively small effect size, and the use of only a single strong control, a peptide with a proline inserted in the place of glycine in one of the already substituted positions. There are a couple of general issues to address and then a number of issues with the data.

1. There is little indication of why zeta was selected as the model and why a screen of all 6 unique TM domains in the complex was not undertaken to arrive at an optimal PITCR. Even if published already, an explanation of the strategy to generate these conditional TM peptides would be helpful. For example, the strategy seems to involve inserting charged residues into the TM domain and generally decreasing the pI. At which point the membrane insertion is suggested to be dependent on lowering the pH and charge. Given that the TCR zeta TM already had a charge in the TM domain (as do all TCR sTM domains) does this make the TCR more or less challenging to engineer in this way? Given that the charges in the TM domain of the TCR are highly significant for their function, does this make the addition of extra charges problematic if the intention is for them to insert into the complex?

2. The modelling of the insertion of the PITCR into a space between the natural zeta and the epsilon complexed to the α subunit. What is the basis of this? Is this resulting in a displacement of zeta from interactions with this copy of epsilon TM? Can figure 9 be redrawn to show the TCR TM configuration with and without the PITCR?

3. The conditions for stimulation appear to be 1 µM OKT3. This is a very high concentration. This might exceed the optimal concentration for cross-linking TCR and may paradoxically be a sub-optimal condition that would be easier to block. Did the authors do a dose-response to assess this?

4. In Figure 3, the data is not convincing as the PITCR-treated condition in A appears very similar to the OKT3 controls in B and C. So the no peptide response to OKT3 in A looks like an outlier. I don't see how the repeats in D could be so tight with this kind of variability in a single experiment.

5. In Figure 5 I have a hard time seeing how the co-localization could be so high. Clearly, there are relatively few strong PITCR signals and lots of additional CD3e signals. So you have a high chance of finding TCR within PITCR. It might also be best to focus on plasma membrane signals as you might expect PITCR to insert into intracellular complexes and this might lead to partial complexes without zeta.

6. In Figure 6, there appears to be some PITCR signal in the supported bilayer. This doesn't surprise me as it's very hard to prevent lipophilic probes from transferring to the bilayer. This suggests to me that the PITCR may be acting just like a membrane probe, which will always show central accumulation in a T cell synapse due to polarization. Therefore, a stronger control is needed. I think a critical control would be a labelled version of the G41P mutated PITCR peptide to ensure that the signal is related to TCR association and not just membrane insertion. It would also be more convincing to show the signal in peripheral microclusters than in the central SMAC, which contains lots of membranes, vesicles, organelles, etc, within the TIRF field. Quantification of the result would also be critical, although results in Figure 5 suggest this will be challenging.

*Reviewer #3 (Recommendations for the authors):*

1. Direct evidence is needed to support the claim that the designed peptides interact with or bind to the TM region of the TCRs. Similarly, evidence should also be provided for mutated peptides that serve as negative controls.

2. Experiments should be better described, including the rationale for the choices of cells.

3. Discussion should be included on how to interpret and consolidate findings from different measurements using different T cells, considering the potential differences between the cell types used.

4. Data or discussion should be included on whether activation by anti-CD3 induces the same allosteric changes as activation by pMHC.

---

## [Author Response]

There are 4 issues that should be addressed:1. Different cells were used in different measurements and the results are combined in the model. More experiments may be needed to link the results from the different experiments.

As we discuss in the comments to the reviewers, the rationale and limitations of using different cell lines are now discussed in the manuscript.

2. Why zeta was selected should be discussed.

We have discussed at length and justified in the comments to reviewers why the zeta chain was selected to create PITCR.

3. Anti-CD3 was used for stimulation in most experiments but the results are discussed in the context of pMHC activation. Does anti-CD3 cause the same allosteric changes as pMHC? If this is an issue, the extrapolation to pMHC should be toned down.

We have toned down in the manuscript, as requested, and discuss that allosteric changes resulting from CD3 activation might not be identical to those in pMHC activation.

4. The hypothesis is that the peptide inhibitor, which resembles the CD3 zeta TM domain and might be expected to displace zeta, actually inhibits allosteric activation. This model really needs structural studies (such as NMR) to confirm. However, we recognize that such studies are likely beyond the scope of the paper, or what could be achieved in a 2-3 month period. The paper would be strengthened if additional data or information could be provided regarding the mechanism of interaction between the peptide and the TCR.

As discussed above, AlphaFold Multimer studies provide a feasible structural model for the binding site of PITCR in the receptor, and additionally offers mechanistic insights, whereby displacement of the zeta chain is part of the inhibitory mechanism of PITCR.

Reviewer #1 (Recommendations for the authors):1. There is little indication of why zeta was selected as the model and why a screen of all 6 unique TM domains in the complex was not undertaken to arrive at an optimal PITCR. Even if published already, an explanation of the strategy to generate these conditional TM peptides would be helpful. For example, the strategy seems to involve inserting charged residues into the TM domain and generally decreasing the pI. At which point the membrane insertion is suggested to be dependent on lowering the pH and charge. Given that the TCR zeta TM already had a charge in the TM domain (as do all TCR sTM domains) does this make the TCR more or less challenging to engineer in this way? Given that the charges in the TM domain of the TCR are highly significant for their function, does this make the addition of extra charges problematic if the intention is for them to insert into the complex?

We thank the reviewer for suggesting to expand the explanation for conditional TM generation; we have expanded the Results section (line 91-94) to provide more information, and we now mention that the acidic residues in the TCR TM allow peptide design with minimal sequence modifications.

The reviewer correctly points out that other TCR transmembrane sequences could potentially have been selected for peptide design, as they contain acidic residues. We initially selected the zeta chain and chose to focus on the peptide generated from its sequence (PITCR) since (i) robust inhibitory effect was observed using multiple techniques, and (ii) it could be used as a model that generates biological insights by allowing us to test the proposed allosteric activation mechanism of TCR. It is possible that peptides derived from other chains would have similar characteristics, but testing this possibility is outside of the scope of this work.

2. The modelling of the insertion of the PITCR into a space between the natural zeta and the epsilon complexed to the α subunit. What is the basis of this? Is this resulting in a displacement of zeta from interactions with this copy of epsilon TM? Can figure 9 be redrawn to show the TCR TM configuration with and without the PITCR?

The reviewers’ comments motivated us to revisit and improve the model of the interaction. We therefore employed AlphaFold-Multimer (AlFoM) to tackle this problem. The updated manuscript significantly improves on the prediction of the binding site to TCR, which is presented in the new Figure 9 and four supplementary figures (Figure 9 —figure supplementary 1-4).

Our first step was to apply AlFoM to the full-length TCR in the absence of added peptide. After an optimization step (described in the Methods section), we were able to obtain an AlFoM model that did an outstanding job predicting the structure of the protein complex; the root mean square deviation between the cryo-EM structure and AlFoM prediction was only 1.27 Å (Figure 9 —figure supplementary 1). Once we validated the approach, we applied AlFoM to predict the binding site of PITCR in the receptor. AlFoM predicted that PITCR docks to the two zeta chains of TCR. Importantly, PITCR binding is predicted to displace both zeta chains, particularly ζ(εγ), which shifts up to 8 Å. Such conformational change can be reasonably expected to hinder allosteric changes that occur during TCR activation. To ensure that the data were robust, we repeated the modeling with the negative control mutant peptide PITCRG41P, which does not inhibit signaling (Figure 3). AlFoM predicts that this inactive peptide binds to the same binding site, but it interacts only weakly with TCR -while six amino acids in PITCR interacted with TCR, only two PITCRG41P residues establish contacts, as shown in Figure 9 —figure supplementary 2-. Similar results were obtained with a second negative control peptide, pHLIP. The data showed that neither of the two peptides caused large displacement of the zeta chains. To further benchmark the approach, we included an additional negative control, the TYPE7 peptide, which targets the unrelated EphA2 receptor. AlFoM predicts that this peptide does not bind to TCR and does not cause a conformational change in TCR.

To summarize, the new AlFoM model not only provides a data-based atomic prediction of the binding site of the peptide in the receptor complex, but additionally provides new insights on the mechanism that underpins the effect of PITCR, namely that it displaces the zeta chain.

3. The conditions for stimulation appear to be 1 µM OKT3. This is a very high concentration. This might exceed the optimal concentration for cross-linking TCR and may paradoxically be a sub-optimal condition that would be easier to block. Did the authors do a dose-response to assess this?

When we used a lower concentration of OKT3, 0.5 µM, we observed less consistent changes in the phosphorylation of Zap70. As a result, we used the higher concentration, which has been used in prior TCR studies. Regardless of the OKT3 concentration used, this approach still represents a somewhat artificial -although widely used- model of TCR activation. However, we are confident about the inhibitory capacity of PITCR, as it inhibited activation of TCR by peptide presentation in antigen-presenting cells (Figure 4), which is a more physiologically-relevant activation mode.

4. In Figure 3, the data is not convincing as the PITCR-treated condition in A appears very similar to the OKT3 controls in B and C. So the no peptide response to OKT3 in A looks like an outlier. I don't see how the repeats in D could be so tight with this kind of variability in a single experiment.

We thank the reviewer for this comment; in retrospect, that data that we presented were not the clearest in representing the effect of PITCR inhibiting calcium signals. We have therefore revised Figure 3 to include a different replicate that more clearly shows the inhibitory role of PITCR.

5. In Figure 5 I have a hard time seeing how the co-localization could be so high. Clearly, there are relatively few strong PITCR signals and lots of additional CD3e signals. So you have a high chance of finding TCR within PITCR. It might also be best to focus on plasma membrane signals as you might expect PITCR to insert into intracellular complexes and this might lead to partial complexes without zeta.

We performed an unbiased analysis of co-localization in the confocal images, which yielded very significant co-localization using the Mander’s coefficient (~0.8-0.9), and Pearson’s (~0.4); maybe this second parameter, contained in Figure 5 Figure supplementary 1, is closer to the reviewer expectation of degree of co-localization. While this study can be informative, co-localization on itself is rarely conclusive. However, the combination between co-localization observed in figures 5 and 6 with the co-IP results make for a compelling case of interaction. It would be indeed useful to identify co-localization in the plasma membrane. However, the close opposition of the plasma membrane with endocytic vesicles severely complicates performing this analysis with light microscopy, since both can be located just a few nm away. To avoid providing biased data, we did not try this approach.

6. In Figure 6, there appears to be some PITCR signal in the supported bilayer. This doesn't surprise me as it's very hard to prevent lipophilic probes from transferring to the bilayer. This suggests to me that the PITCR may be acting just like a membrane probe, which will always show central accumulation in a T cell synapse due to polarization. Therefore, a stronger control is needed. I think a critical control would be a labelled version of the G41P mutated PITCR peptide to ensure that the signal is related to TCR association and not just membrane insertion. It would also be more convincing to show the signal in peripheral microclusters than in the central SMAC, which contains lots of membranes, vesicles, organelles, etc, within the TIRF field. Quantification of the result would also be critical, although results in Figure 5 suggest this will be challenging.

We do not agree that the PITCR that appears to be cSMAC-localized is in the supported bilayer. We are using very low pMHC densities so there is essentially no drag on the bilayer itself to push PITCR to a central location. Furthermore, even PITCR in the plasma membrane would not be expected to be driven centrally. We have shown that pMHC:TCR is uniquely strongly coupled to the cytoskeleton and that even other large molecules (like LFA) are not dragged to the c-SMAC. Please see, for example Proc. Natl. Acad. Sci. USA, 2009, 106, 31, 12729-12734: “Cluster size regulates protein sorting in the immunological synapse” , where we show that it takes large-scale clustering of other molecules to render them competitive with pMHC:TCR for c-SMAC localization. While we do not claim that our observations are proof, on their own, that PITCR is TCR associated, the c-SMAC localization is a piece of evidence that PITCR is feeling a force that is rather selectively transmitted to the pMHC:TCR complex. We also know, from many of our published experiments, that centralized c-SMAC driving forces do not non-specifically couple to molecules including lipid probes in either the supported membrane or in the T cell membrane itself.

Reviewer #3 (Recommendations for the authors):1. Direct evidence is needed to support the claim that the designed peptides interact with or bind to the TM region of the TCRs. Similarly, evidence should also be provided for mutated peptides that serve as negative controls.

As we discuss at length in the second response to reviewer 1, new AlphaFold Multimer data presented in Figure 9 strongly supports the claim that PITCR binds to the transmembrane region of TCR. Three different peptides are successfully used as negative controls in the AlphaFold Multimer prediction.

2. Experiments should be better described, including the rationale for the choices of cells.

The updated manuscript contains discussion about the rationale for the selection of cells.

3. Discussion should be included on how to interpret and consolidate findings from different measurements using different T cells, considering the potential differences between the cell types used.

As the reviewer suggest, we have updated the manuscript in pages 17 and 18 to include discussion on the particularities of the use of the different T cells.

4. Data or discussion should be included on whether activation by anti-CD3 induces the same allosteric changes as activation by pMHC.

We have updated the manuscript to discuss that differences might exist in TCR activation by OKT3 and pMHC, as suggested by the reviewer. Figure 8 shows that the levels of co-IP in the presence of detergent are altered by OKT3 activation of TCR. It has recently been established (PMID: 34260912) that this assay allows investigation of allosteric changes that contribute to the activation of TCR. We think that this evidence is supportive of allosterism in TCR activation. Additionally, the TCR proximal signaling is similar when Jurkat T cells are activated by OKT3 and when TCR is activated by pMHC. We can reasonably argue that the peptide acts similarly in both conditions, since the peptide also exerts an inhibitory effect in T cells activated by antigen-presenting cells (Figure 4).